# LIMIT: Language Identification, Misidentification, and Translation using Hierarchical Models in 350+ Languages

**Milind Agarwal**      **Md Mahfuz Ibn Alam**      **Antonios Anastasopoulos**

Department of Computer Science, George Mason University

{magarwa, malam21, antonis}@gmu.edu

## Abstract

Knowing the language of an input text/audio is a necessary first step for using almost every NLP tool such as taggers, parsers, or translation systems. Language identification is a well-studied problem, sometimes even considered *solved*; in reality, due to lack of data and computational challenges, current systems cannot accurately identify most of the world's 7000 languages. To tackle this bottleneck, we first compile a corpus, MCS-350, of 50K multilingual and parallel children's stories in 350+ languages. MCS-350 can serve as a benchmark for language identification of short texts and for 1400+ new translation directions in low-resource Indian and African languages. Second, we propose a novel *misprediction*-resolution hierarchical model, LIMIT, for language identification that reduces error by 55% (from 0.71 to 0.32) on our compiled children's stories dataset and by 40% (from 0.23 to 0.14) on the FLORES-200 benchmark. Our method can expand language identification coverage into low-resource languages by relying solely on systemic misprediction patterns, bypassing the need to retrain large models from scratch.[1]

## 1 Introduction

Building natural language processing (NLP) tools like machine translation, language identification, part of speech (POS) taggers, etc. increasingly requires more and more data and computational resources. To attain good performance on a large number of languages, model complexity and data quantity must be increased. However, for a majority of the world's 7000 languages, large amounts of data are often unavailable which creates a high barrier of entry (Blasi et al., 2022; Joshi et al., 2020; Khanuja et al., 2023). Increasing model complexity for large-scale models also requires disproportion-

---

[1]Data, code, and models are publicly available on GitHub under permissive licenses. Repository: https://github.com/magarw/limit

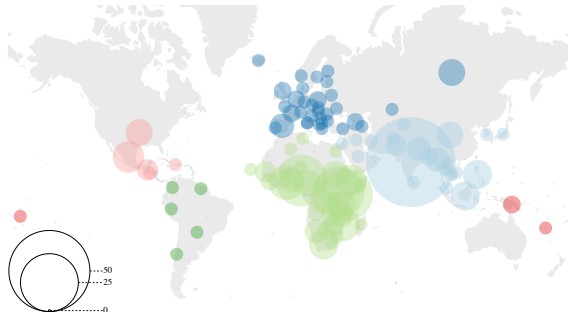

Figure 1: Most languages in our dataset are from the Indian Subcontinent and Sub-Saharan Africa, with significant minorities from Europe (primarily in the role of the high-resource language parallel translation available for each story). Color broadly indicates continent or region (North America, South America, Africa, Europe, Asia, Oceania) and size indicates number of languages per country in our dataset.

ate amount of computational resources, further disincentivizing researchers to work towards including these languages in modern NLP systems.

A popular data collection approach is large-scale web mining (Tiedemann and Nygaard, 2004; Bañón et al., 2020; Schwenk et al., 2021b), where large parts of the internet are scoured to find training data for data-hungry NLP algorithms. When faced with a sentence or phrase, such algorithms must know how to reliably sort this text into the appropriate language bucket. Since the web is replete with content in a variety of languages, a model needs to recognize text in a sufficiently large number of these languages with high accuracy. Identifying parallel bitext is even more demanding as a machine translation system must also be available to correctly identify and align parallel data (Vegi et al., 2022; Kunchukuttan et al., 2018). This data-collection paradigm becomes inaccessible for low-resource languages because high-quality translation models usually require substantial amounts of parallel data for training, which is often unavailable. Without high-quality language identification and

translation system, it becomes practically impossible to mine the internet for relevant text during such collection efforts. Additionally, mispredictions by language identification and data collection algorithms can increase inter-class noise, reducing the crawled data's quality, and harming performance in downstream tasks without strong quality evaluation metrics (Kocyigit et al., 2022).

How can we address these challenges and build high-quality identification and translation for low-resource languages?

**Resource Creation**    Highlighting the need for resource creation in low-resource languages, we first share a new parallel children's stories dataset, MCS-350, created using two resources: African Storybooks Initiative[2] and Indian non-profit publishing outfit Pratham Books' digital repository Storyweaver[3] (available under permissive Creative Commons licenses). The combined dataset includes original and human-translated parallel stories in over 350 languages (visualized in Figure 1) and we merge, preprocess, and structure it so it is easily utilizable by NLP researchers for training and benchmarking (§2).

**Machine Translation**    Armed with parallel stories in many low-resource African and Indian languages, we tackle machine translation in resource-constrained situations next. If we aim to collect parallel data in low-resource languages, language identification itself is insufficient and we need high-quality translation models as well. We utilize a pre-trained multilingual translation model (Alam and Anastasopoulos, 2022) and explore training with hierarchical language-level and language family-level adapter units to translate children's stories at the page level (§3).

**Language Identification**    Finally, we take on the biggest bottleneck in low-resource language data collection efforts - language identification. We propose LIMIT - a misidentification-based hierarchical modeling approach for language identification, that utilizes data and computational resources efficiently and shows cross-domain generalization. The proposed approach is exciting because unlike previously published language identification models like AfroLID (Adebara et al., 2022), CLD3 (Salcianu et al., 2020) and Franc[4], LIMIT avoids train-

[2] https://www.africanstorybook.org/
[3] https://storyweaver.org.in/
[4] https://github.com/wooorm/franc/

| Family | Languages | Sentences |
|---|---|---|
| Niger-Congo | 129 | 142605 |
| Indo-European | 84 | 169823 |
| Nilo-Saharan | 22 | 23204 |
| Sino-Tibetan | 21 | 19264 |
| Austronesian | 18 | 28096 |
| Afro-Asiatic | 15 | 20266 |
| Dravidian | 13 | 35638 |
| Austro-Asiatic | 10 | 22989 |

Table 1: Our compiled dataset MCS-350 contains stories from a diverse set of languages families, mostly coming from Africa and India. Prominent language families with with 20K+ sentences across languages shown.

ing large multilingual models for a new set of languages and still outperforms existing systems. Large multilingual models often require thousands of sentences for training, ex. AfroLID (Adebara et al., 2022) collects and trains on over 4000 sentences per language. On the other hand, for many low-resource languages in India and Africa, we may not even be able to collect 1000 sentences at first 2. Also, in contrast with other recent work in hierarchical language identification (Goutte et al., 2014; Lui et al., 2014; Bestgen, 2017; Jauhiainen et al., 2019), our work stands out because it accounts for *mispredictions* made by existing trained models. Unlike other work, it does not predict a group/language family first, but rather directly learns confusion relationships between language pairs (which may not be from the same language family). By leveraging hierarchically organized units on top of a root model, we avoid complete retraining, saving computational resources, while increasing coverage into many new and understudied languages and language pairs (especially those between two low-resource languages) (§4).

To summarize, our main contributions are:

1. We compile MCS-350, a dataset of 50K+ parallel children's stories from African Storybooks Initiative and Storyweaver in 350+ languages (§2).
2. We share a machine translation benchmark enabling translation evaluation in more than 1400 new translation directions (§3).
3. We propose LIMIT, a misidentification-based hierarchical model, that can use limited data to better identify low-resource languages (§4).

| Dataset | New languages | New pairs |
|---------|---------------|-----------|
| Microsoft | 67 | 2835 |
| FLORES-200 | 51 | 1449 |
| OPUS | 82 | 2853 |

Table 2: MCS-350 enables MT evaluation between 1400+ new pairs compared to existing benchmarks.

| Script | Languages | Examples |
|--------|-----------|----------|
| Devanagari | 38 | Hindi, Marathi |
| Cyrillic | 14 | Russian, Bulgarian |
| Arabic | 8 | Arabic, Persian |
| Tibetan | 3 | Tibetan, Ladakhi |
| Telugu | 3 | Telugu, Konda |
| Odia | 3 | Odia, Ho, Kui |

Table 3: Our dataset contains stories in many writing systems other than Latin, especially those from the Indian Subcontinent. Prominent non-Latin writing systems in MCS-350 are shown above.

## 2   MCS-350 Data Curation

We identify two large-scale parallel repositories - African Storybooks Initiative and Pratham Books' Storyweaver, both under permissive Creative Commons Licenses, with their storybooks available for non-commercial and research use. African Storybooks Initiative hosts parallel translated and human-verified children's stories in over 200 African languages. Pratham Books is a non-profit Indian publisher that aims to increase literacy of children and adults alike in Indian languages. Their digital repository, Storyweaver, publishes parallel translated stories in 300+ languages. This includes not only Indian languages but also African, European, and Indigenous languages from the Americas.

### 2.1   Parallel Dataset

We collect stories through a mix of web scraping and public APIs, preprocess them to remove mismatched/incorrect text, extract monolingual text for language identification and parallel text for machine translation. We maintain metadata about authors, translators, illustrators, reading level, parallel translations, and copyrights for each story. We remove stories that are either empty or those from non-English languages that have over 50% pages containing majority English text with 90% confidence using langdetect (Nakatani, 2010). This leaves us with ~52K stories.

Note that both African Storybooks Initiative and Pratham Storyweaver human verify stories and language. However, there are several abandoned translation projects and completed but unverified stories that need automated checking. Therefore, our preprocessing is meant for unverified stories, and may introduce noise in the collected data. By improving the preprocessing filters, we can likely further improve the quality of the unverified stories in the corpus. Collected stories in the pre-merge stage are available with their associated metadata in the repository.

### 2.2   Multilingual Documents

MCS-350 contains multilingual stories with language identifiers denoted by $L_1\_L_2$ for a story multilingual in $L_1$ and $L_2$. Such stories include text in multiple languages within the same page. Text may be code-mixed or consecutively presented. To extract as many parallel sentences as possible to support vulnerable languages and also create new translation directions, we employ string-similarity based matching to identify the segments corresponding to the high-resource language in the pair, and therefore automatically generating parallel sentences from 10K pages across 52 languages. E.g., through this process, we extracted 1000+ sentences in Kui (0 sentences pre-extraction), a minority Dravidian language with about 900K native speakers. We manually verified all extracted monolingual text after using string matching on multilingual stories.

### 2.3   Language Varieties/Lects

We attempt to separate language varieties/lects into unique prediction classes if there is sufficient training data for them ($\geq$ 1000 sentences). If an ISO code is unavailable for the lect, we assign a class name with the ISO code and the subdivision specified as: ISO_SUBDIVISION. For instance, we separated Gondi's South Bastar lect (GON_BASTAR, 4000+ sentences) from the generic language code for Gondi (GON). For fair evaluation and comparison, we provide manual mappings for any non-standard identifiers from the output space of various language identification tools. Lects with too little data are merged into their parent language, e.g., "Bangla (Bangladesh)" merged into "Bengali".

### 2.4   Data Overview

MCS-350 covers over 350 languages from a diverse pool of language families. In Table 1, we share the

| Model | Avg$_{\text{ALL}}$ | Avg$_{\text{AFRI}\to\text{AFRI}}$ | Avg$_{\text{X}\to\text{ENG}}$ | Avg$_{\text{ENG}\to\text{X}}$ | Avg$_{\text{Y}\to\text{FRA}}$ | Avg$_{\text{FRA}\to\text{Y}}$ |
|---|---|---|---|---|---|---|
| Baseline | 11.87 | 10.19 | 18.79 | 13.20 | 15.64 | 12.55 |
|  | (6.31) | (5.06) | (7.75) | (8.19) | (5.22) | (5.81) |
| L-Fine | 19.52 | 18.21 | 30.38 | 17.46 | 21.93 | 17.86 |
|  | (10.33) | (10.06) | (13.63) | (8.46) | (4.87) | (6.86) |
| F-Fine | **24.93** | **23.58** | **35.66** | **25.26** | **27.06** | **21.36** |
|  | (11.74) | (11.31) | (14.36) | (13.72) | (6.00) | (7.32) |
| Unique Pairs | 88 | 58 | 16 | 16 | 14 | 14 |

Table 4: spBLEU across 176 translation directions involving African languages, we see that including phylogenetic information helps in translation, with the family-based F-Fine model showing the best performance, on average. **Avg**$_{\text{AFRI}\to\text{AFRI}}$ denotes the overall average spBLEU of translation between two African languages. **Avg**$_{\text{X/Y}\to\text{ENG/FRA}}$ and **Avg**$_{\text{ENG/FRA}\to\text{X/Y}}$ denote translating into and out of English/French respectively. Parentheses below the averages represent standard deviations. Baseline refers to a DeltaLM model finetuned on 26 languages without adapters. We can see that it is harder to translate out of English than into English.

| Lang Pair | $\Delta$ | Lang Pair | $\Delta$ |
|---|---|---|---|
| ENG-XHO | 20.1 | ENG-HAU | 18.8 |
| FRA-LUG | 3.6 | NSO-LUG | 3.0 |
| LUG-KIN | 2.9 | KIN-LUG | 2.4 |
| NYA-LUG | 2.1 | ENG-KAM | 1.8 |
| IBO-LUG | 1.7 | ENG-LUG | 1.5 |
| ZUL-LUG | 1.5 | FRA-TSO | 1.3 |
| XHO-LUG | 1.2 | FRA-YOR | 1.1 |
| NSO-TSO | 1.0 | AMH-LUG | 1.0 |

Table 5: Despite MCS-350 and FLORES-200 having widely different domains, several translation directions see cross-domain improvements. $\Delta$ indicates spBLEU improvements in the F-Fine model over the Baseline

number of languages and the number of sentences in each language family in the dataset. The data is roughly evenly split between stories from the large Niger-Congo and Indo-European language families, with a sizeable minority in other language families like Nilo-Saharan, Sino-Tibetan, Austronesian, Dravidian, Creole, etc. About 70% of the dataset's languages use the Latin script or its extended variants with diacritics. However, the data is still quite typographically rich, and stories with non-Latin scripts are in abundance, enumerated in Table 3.

Compared to highly multilingual translation benchmarks like NTREX (parallel data of 128 languages; Federmann et al., 2022), FLORES-200 ($n$-way, 200 languages; NLLB Team et al., 2022), or OPUS-100 (parallel data for 99 languages to/from English; Aharoni et al., 2019), our benchmark introduces up to 82 new languages leading to more than 1400 new language pairs (see Table 2).

## 3 Machine Translation Benchmark

While it is true that resource creation in low-resource languages requires fine-grained and high-quality language identification, collecting *parallel* data additionally requires high-quality MT (§1). In this section, we explore phylogeny-based hierarchical adapter units to improve translation quality between two African languages, and between African languages and English/French.

### 3.1 Data

We exploit the parallel nature of children's stories in MCS-350 and ensure that all training stories are separate from test (1000 pages) stories. This is done to get a more realistic estimate of translation quality on new stories. For languages with < 1000 pages across stories, we use 500-page test sets.

### 3.2 Experimental Settings

As our baseline, we used the model from Alam and Anastasopoulos (2022), which is the best-performing publicly available model from the WMT Shared Task on Large Scale Evaluation for African Languages (Adelani et al., 2022).[6] They first fine-tuned the DeltaLM[7] model (Ma et al., 2021) in 26 languages. After that, they added lightweight language-specific adapter layers (Pfeiffer et al., 2022) and fine-tuned only the adapters in those 26 languages. We can either use a single adapter per language (L-Fine) or organize the adapters in a phylogenetically-informed hierarchy

---

[6]Ranked third in the Shared Task. Top two systems were industry submissions that are not publicly available.

[7]https://aka.ms/deltalm

| Model | $F_1$ | Supported | Common | Total (with LIMIT) |
|---|---|---|---|---|
| CLD3 (Salcianu et al., 2020) | 0.11 | 101 | 81 | 376 |
| langid.py (Lui and Baldwin, 2012) | 0.09 | 97 | 73 | 380 |
| Franc[5] | **0.18** | **369** | 116 | **609** |
| fastText (Joulin et al., 2017) | 0.10 | 176 | 117 | 415 |
| HeLI-OTS (Jauhiainen et al., 2022a) | 0.13 | 200 | 81 | 475 |

Table 6: This table shows different popular language identification ssytems, their $F_1$ scores on MCS-350, supported languages, common languages, and total coverage with LIMIT. Franc, trained on UDHR data, outperforms other systems both on performance and coverage, and will serve as the root model for our experiments. Macro $F_1$ score is computed across all 355+ languages to identify a system with the best overall coverage and accuracy.

(F-Fine) so that similar languages share language-family and genus-level adapters (Faisal and Anastasopoulos, 2022). We perform both L-Fine and F-Fine experiments using the publicly available code [8] and also share an additional baseline by finetuning the DeltaLM model without adapters. Details on phylogenetic trees and reproducibility are in Appendix §A.3.

### 3.3 Evaluation

In Table 4, we show the performance of our L-Fine and F-Fine models compared to the baseline on our test set. We evaluate using three well-known MT metrics: BLEU (Papineni et al., 2002), CHRF++ (Popović, 2017), and spBLEU (NLLB Team et al., 2022). For spBLEU, we use the FLORES200 SPM model to create subwords.

Based on all three metrics, our L-Fine model outperforms the Baseline model consistently by 4.0-11.5 spBLEU points by just fine-tuning with language-specific adapters. Our F-Fine model outperforms the L-Fine model by 5.0-7.5 spBLEu points by fine-tuning only some shared parameters among languages and language-specific adapters. We also test our models on a public benchmark, FLORES200 (Appendix §B), and observe that due to the domain shift, L-Fine and F-Fine models under-perform the Baseline.

Despite this domain shift, several low-resource language pairs benefit from adapter fine-tuning across domains. We report these language pairs and their respective spBLEU gains for the F-Fine model in Table 5. We get the highest gains for English-Xhosa (20.1 points) and English-Hausa (18.8 points) across domains, both of which had poor performance from the Baseline model with spBLEU of 3.5 and 4.5, respectively. We also no-

tice cross-domain improvement in some translation directions involving two African languages such as Ganda-Kinyarwanda (2.9 points) and Northern Sotho-Ganda (3.0 points). Exhaustive results for other language pairs can be found in Appendix §B.

## 4 Language (Mis)Identification Benchmark

Language identification (LID) affects low-resource language resource creation efforts severely (Jauhiainen et al., 2019; Schwenk et al., 2021a) because to collect data, we need accurate language identifiers that themselves need high-quality data to train(Burchell et al., 2023) , creating a vicious cycle. Low-quality systems often make mispredictions which increases inter-class noise and reduces the crawled data's quality (Kocyigit et al., 2022; Burchell et al., 2023) both for the predicted language and the true language. To correct mispredictions and improve accuracy in supported languages with limited data, we propose a hierarchical modeling approach.

Hierarchical modeling is an extremely popular choice for a wide variety of algorithmic tasks and it has been explored for language identification as well (Goutte et al., 2014; Lui et al., 2014; Bestgen, 2017; Jauhiainen et al., 2019). However, previous work has focused on predicting language group/family first, followed by finer-grained predictions with a smaller set of classes. Our work departs from this paradigm in two ways - first, we bring focus onto expanding language identification coverage in pre-trained or off-the-shelf systems without retraining, and second, we predict a prior and posterior language based on confusion and misprediction patterns of the model directly (without predicting language family/group first).

Under our technique, we first choose a well-performing root model with high-coverage that

---

[8]https://github.com/mahfuzibnalam/large-scale_MT_African_languages

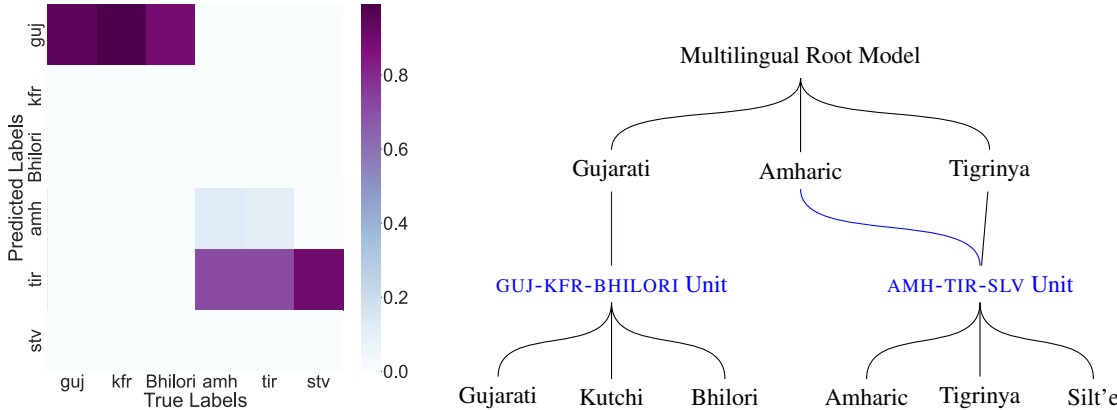

Figure 2: Subset of the multilingual `root` model's confusion matrix (6 languages). Using the confusion matrix, clusters of highly confused languages are identified and confusion-resolution units trained according to the tree shown on the right. The tree, for demonstration purposes, is a subset of the entire tree which has 9 confusion-resolution units

provides us with the base/prior prediction. Such base predictions are obtained for a sample of `MCS-350`'s training set, allowing us to identify systemic confusion patterns embedded within the model using a confusion matrix. Based on the identified misprediction patterns (which may or may not be between languages in the same family), we train lightweight confusion-resolution subunits that can be attached onto the `root` model to make the posterior prediction. Our results show that, with this architecture, a small sample of data is sufficient to investigate pretrained, off-the-shelf, or blackbox commercial models and identify systemic misprediction patterns across domains.

## 4.1 Experimental Settings

**Wide-Coverage `root` Model**    To pick an appropriate `root` model to test our misidentification-based hierarchical approach, we compare several state-of-the-art pre-trained models (§4.2) and choose the system with the highest macro-$F_1$ score, giving equal importance to all languages in `MCS-350`.

**Traditional Hierarchical group-first Model** Classical hierarchical models predict language family/group first, followed by the specific language (Goutte et al., 2014; Lui et al., 2014; Bestgen, 2017; Jauhiainen et al., 2019). These groups often have phylogenetic backing and are not *learned* through the output distribution of the root model. For benchmarking, we train this traditional hierarchical `group` model as well (Table 7).

**N-Way Multilingual `multi` Model**    To contrast our work with typical large multilingual modeling where there is no architectural class hierarchy, we train a large fastText multilingual model with all 350+ languages (`multi`). With a large number of classes, we know that low-resource languages suffer due to class imbalance, even with upsampling. But, we still include performance results from `multi` in Table 7 to compare it with the two hierarchical approaches.

**LIMIT's Confusion-resolution Units**    We use fastText (Joulin et al., 2017) to train small models that will specialize in distinguishing between 2-3 highly-confused languages each.[9] Up to 1000 sentences/language are used for training, and 100 randomly selected sentences across stories are reserved as the final test set. We train our own embeddings because existing multilingual embeddings (Devlin et al., 2019) are not trained on sufficiently wide low-resource language data.

**Evaluation Metric**    To select a `root` model, performance is compared based on aggregated macro-$F_1$ scores across languages (Table 6). To compare the performance of the `root` model and LIMIT (our proposed approach) on `MCS-350`, our benchmark dataset, and the existing FLORES-200 benchmark, we report language-level $F_1$ scores (Table 7).

---

[9]embedding dim= 100, learning rate= 0.5, loss= negative sampling. We explored several starting learning rates and settled on lr=0.5 due to optimal performance on a small sampled dev set. FastText's lrUpdate parameter, by default, reduces the starting learning rate gradually over epochs. We don't optimize for embedding dimension and use fastText's default, 100, for supervised training.

| Lang | MCS-350 (LIMIT's Domain) | | | | FLORES-200 (Out-of-Domain) | | | |
|---|---|---|---|---|---|---|---|---|
| | root | multi | group | **LIMIT** | root | multi | group | **LIMIT** |
| GUJ | 0.49 | 0.44 | 0.58 | **0.63** | 1.00 | 0.81 | 0.99 | 0.99 |
| └KFR | | 0.78 | 0.83 | **0.80** | | | | |
| └BHIL | | 0.63 | 0.03 | **0.28** | | | | |
| AMH | 0.20 | 0.81 | 0.78 | **0.83** | 0.60 | 0.95 | 0.93 | **0.99** |
| TIR | 0.56 | 0.93 | 0.94 | **0.85** | **0.99** | 0.96 | 0.95 | 0.95 |
| └STV | | 0 | 0 | 0.00 | | | | |
| BEN | 0.47 | 0.82 | 0.05 | **0.85** | 1.00 | 0.94 | 0.96 | 0.99 |
| └ASM | | 0.63 | 0.89 | **0.89** | 0.00 | 0.98 | 0.96 | **0.66** |
| └CDZ | | 0.57 | 0.87 | **0.93** | | | | |
| ZHO | 0.61 | 0 | 0 | **0.68** | 0.99 | 0 | 0.01 | 0.99 |
| └YUE | | 0 | 0.02 | **0.14** | 0.00 | 0 | 0 | 0.00 |
| TEL | 0.92 | 0.75 | 0.80 | **0.94** | 1.00 | 0.83 | 0.91 | 1.00 |
| └KFC | | 0.66 | 0.66 | **0.66** | | | | |
| KAN | 0.70 | 0.77 | 0.78 | **0.81** | 1.00 | 0.96 | 0.93 | 0.99 |
| └KFA | | 0.66 | 0.68 | **0.52** | | | | |
| TSO | 0.49 | 0.53 | 0.34 | **0.67** | 0.97 | 0.79 | 0.72 | 0.97 |
| └TSC | | 0.74 | 0.52 | **0.77** | | | | |
| DAGAARE | 0.83 | 0.48 | 0.54 | **0.87** | | | | |
| └MZM | | 0.69 | 0.88 | **0.82** | | | | |
| KAT | 0.66 | 0.46 | 0.42 | **0.80** | 1.00 | 0.72 | 0.70 | 1.00 |
| └BBL | | 0.82 | 0.66 | **0.66** | | | | |
| AVG | 0.29 | 0.56 | 0.47 | **0.68** | 0.77 | 0.70 | 0.73 | **0.86** |

Table 7: LIMIT improves $F_1$ scores over the root, multi, and group models on both our children's stories dataset and out-of-domain FLORES-200. The traditional hierarchical approach group underperforms the multilingual model multi on both MCS-350 and FLORES-200. Empty entries indicate unsupported languages and bolded entries indicate noteworthy differences in $F_1$ scores. Nested languages are misidentified as the parent in root. Note that for FLORES-200, the root model gets 0 $F_1$ score on ASM and YUE but both languages are covered by the dataset.

## 4.2 Pre-trained root Models

In Table 6, we show macro-$F_1$ scores across all 350+ languages for popular pretrained identification systems like Google's CLD3, Langid.py, Franc, fastText (Joulin et al., 2017), and HeLI-OTS (Jauhiainen et al., 2022a). Franc, built using the Universal Declaration of Human Rights (UDHR) data, comes out with the best macro-$F_1$, covering 30% of our languages (105/356 languages). It is derived from guess-language[10] which uses a mix of writing system detection and character-level trigrams. Hence, we use Franc as the root system for our misprediction-based hierarchical modeling experiments. The overall low scores on human-

written sentences in MCS-350 (all systems achieve an $F_1$ score < 0.20) are worth noting, and indicate that off-the-shelf systems ultimately tend to perform really well only on some languages, despite officially supporting hundreds of languages.

## 4.3 Language (Mis)identification

Next, we inspect the best-performing root model's confusion matrix on MCS-350's training set (a representative example is shown in Figure 2) to understand and identify misprediction patterns. For each test language, we divide the root model's predictions by the total number of tested examples giving us a hit ratio for each pair. E.g., (Gujarati, Kutchi) would represent the ratio of Kutchi sentences that were misidentified

---

[10] https://github.com/kent37/guess-language

as Gujarati. Upon inspection of the confusion matrix, we identified the following 9 clusters with a high confusion ratio (> 0.7). According to our experimental approach outlined in §4.1, we train a lean fastText classifier for each of these clusters, that will specialize in differentiating between these highly-confused languages:

1. Gujarati, Kutchi, Bhilori
2. Amharic, Tigrinya, Silt'e
3. Koda, Bengali, Assamese
4. Mandarin, Yue Chinese
5. Konda, Telugu
6. Kodava, Kannada
7. Tsonga, Tswa
8. Dagaare, Mumuye
9. Bats, Georgian

## 4.4 Expanded Language Coverage

We report $F_1$ scores for each of the 9 highly confused clusters' languages (Table 7) and observe that languages in each cluster share writing systems and are often phylogenetically related. Our misidentification-based model, LIMIT, is successful at improving $F_1$ scores on both our newly collected MCS-350 dataset as well as the public benchmark, FLORES-200. On MCS-350, LIMIT improves $F_1$ scores from 0.29 to 0.68, a 55% error reduction. Of the multidomain data available in FLORES-200 (11/21 languages), LIMIT improves $F_1$ from 0.77 to 0.86, a 40% error reduction, demonstrating that our method's utility is not restricted to the training data's domain.

Note that hierarchical modeling could be viewed as further complicating a simple root model, but we contend that this is valuable when retraining is not an option due to lack of data, closed-source code, etc (Section 5). This simple extension allows us to extend a high-coverage root model to newer languages or domains that have small amounts of training data, while maintaining high-quality predictions. Furthermore, our hierarchical method LIMIT also outperforms a system multi that is trained on all the languages in the test set.

## 4.5 Sentence Length and Domain

For several languages like Gujarati, Amharic, Bengali, and Mandarin, low $F_1$ scores for MCS-350 compared to high $F_1$ scores on FLORES-200 indicate that shorter texts in the children's stories domain are much harder to identify. This is expected due to limited feature signals in shorter texts but it

is worth noting that that is the opposite of our findings in the machine translation task (§3.3), where translating shorter texts in MCS-350 proved easier than translating FLORES-200 data. Our *misprediction*-based hierarchical is not only easier to train with limited data, but also brings valuable cross-domain language identification improvements.

## 5 Related Work

### 5.1 Parallel Datasets

Language identification models tend to use popular training datasets like UDHR (Vatanen et al., 2010), Blodgett et al. (2017) for social media, King and Abney (2013) (web-crawl in 30 languages), FLORES-200, JW-300 (Agić and Vulić, 2019) (multilingual articles from Jehovah's Witness' website) etc.

A recently published dataset, BLOOM (Leong et al., 2022), leverages text and audio in children's stories from similar sources (African Storybooks, The Asia Foundation, Little Zebra Books etc.) to create benchmarks for image captioning and speech recognition. However, their data is monolingual, unaligned, and cannot be used for machine translation. We leveraged the highly parallel nature of the collected storybooks, five times the number of stories in BLOOM, and created test sets and baselines for understudied translation directions.

It is also important to us to avoid representation washing (Caswell et al., 2020) and we clearly highlight the sources of noise from unverified stories in our merged dataset. With stricter preprocessing filters applied at the pre-merge stage, a 'cleaner' dataset could be produced, like in Burchell et al. (2023). We provide access to our data at all such timesteps in the preprocessing pipeline so researchers are not required to use the final dataset, but may use an earlier raw version and preprocess it themselves according to their needs.

### 5.2 Machine Translation

Thousands of languages are spoken worldwide, so representing them with bilingual models would require thousands of models. Neither scalability nor adaptability makes this an ideal solution. Through various training methods (Aharoni et al., 2019; Wang et al., 2020), model structures (Wang et al., 2018; Zhang et al., 2021), and data augmentation (Tan et al., 2019; Pan et al., 2021) a variety of research has attempted to improve multilingual translation models. Adapter units were initially pro-

posed for light-weight domain adaptation (Vilar, 2018) and then also for extending large pre-trained models to a downstream tasks and using bilingual adapters (Houlsby et al., 2019; Bapna and Firat, 2019).

### 5.3 Language Identification

Text-based language identification is usually modelled as a classification task. By increasing the number of languages a classifier must predict, average accuracy generally tends to decrease (Jauhiainen et al., 2017), a problem we propose to tackle by leveraging a misprediction-based hierarchical approach. To distinguish between closely related languages, a lot of exciting research has been published at various editions of VarDial - The Workshop on NLP for Similar Languages, Varieties and Dialects (Aepli et al., 2022; Scherrer et al., 2022; Chakravarthi et al., 2021; Zampieri et al., 2020, 2014).

Over the last 3 iterations of VarDial from 2019-2022, many new datasets and techniques to identify Romance languages (Jauhiainen et al., 2022b; Zaharia et al., 2021), Nordic languages (Haas and Derczynski, 2021), Uralic languages (Jauhiainen et al., 2020), German lects (Mihaela et al., 2021; Siewert et al., 2020), and the Slavic language continuum (Popović et al., 2020; Abdullah et al., 2020) were published. In contrast, we see only a handful papers and tasks on Indian languages at the venue with 2 focusing on Indo-Aryan and 2 focusing on Dravidian languages (Nath et al., 2022; Bhatia et al., 2021; Jauhiainen et al., 2021; Chakravarthi et al., 2020), and no papers or tasks, to our knowledge, on African languages. Outside the venue, recently published models like AfroLID (Adebara et al., 2022) for language identification and IndicTrans2 (AI4Bharat et al., 2023) for Indic-language translation are great large-scale efforts in the low-resource language space.

Brown (2014), a notable technique, trains richer embeddings with non-linear mappings and achieves substantial improvements in downstream language identification on 1400+ languages. However, we do not benchmark with this technique because the paper does not contain any experiments in low-resource training setups. Training data is about 2.5 million bytes/language, while we are working with <50K bytes/language. Therefore, exploring non-linear embedding mappings in low-resource settings (Brown, 2014) is left for future work.

### 5.4 Hierarchical Modeling

Hierarchical approaches have proved successful in solving a myriad of computational problems, and have proved useful in language identification previously. The widely used approach first predicts a preliminary language group/family, and then another fine-tuned prediction from a smaller set of output classes contained within the language group/family (Goutte et al., 2014; Lui et al., 2014; Bestgen, 2017; Jauhiainen et al., 2019). In contrast, our work extends architecture to account for mispredictions made by existing trained models, and does not predict a group/language family first, but rather directly learns confusion relationships between language pairs. Then, similar to Bestgen (2017); Goutte et al. (2014), we train smaller classifiers for a fine-tuned posterior prediction. However, our approach departs from their paradigm in that our classifiers may also distinguish between highly-confused languages which belong to different language families.

## 6 Conclusion

In this work, we tackle the lack of resources for many of the world's languages and release a large, massively parallel children's stories dataset, `MCS-350`, covering languages from diverse language families, writing systems, and reading levels. Since translation is crucial for parallel resource creation, we explore adapter-based networks fine-tuned on a phylogenetic architecture, and utilize `MCS-350` to create new translation benchmarks for vulnerable and low-resource languages. We demonstrate large improvements in the children's story domain and cross-domain improvement for several language pairs (on the FLORES benchmark dataset). On the algorithmic front, we introduce `LIMIT`, a hierarchical, misprediction-based approach to counter the inaccuracies of pre-trained language identification systems. Our method increases language coverage and prediction accuracy bypassing complete retraining, and shows cross-domain generalization despite being trained on our `MCS-350` dataset.

In the future, we hope to further investigate misprediction-based hierarchical language identification across more datasets, with more configurations, and extensions such as probabilistic branching, automated constructions etc. As a natural next step, we will utilize `LIMIT` in a web-crawl to find and collect more low-resource language data.

## Limitations

Our dataset covers 350+ text-based languages. However, out of the 7000 languages in the world, many are primarily spoken languages and do not have a presence in the form of articles, textbooks, stories etc. Therefore, language identification for speech is crucial and we plan on extending our text-based work to speech in future work.

While our proposed method LIMIT shows cross-domain improvements, we acknowledge that our system, like other language identification systems, is not perfect and may still make classification errors on new domains, text lengths, or orthographies. We encourage researchers to keep this in mind when applying our proposed method to their work.

## Ethics Statement

Data used, compiled, and preprocessed in this project is freely available online under Creative Commons licenses (CC BY 4.0). Stories from the African Storybooks Initiative (ASI) are openly licensed, can be used without asking for permission, and without paying any fees. We acknowledge the writers, authors, translators, illustrators of each of the books and the ASI team for creating such a valuable repository of parallel storybooks in African languages. Stories from the Pratham Storybooks' Storyweaver portal are available under open licensing as well, and we preserve metadata for the author, illustrator, translator (where applicable), publisher, copyright information, and donor/funder for each book, in accordance with Storyweaver's guidelines. Since stories hosted on African Storybooks Initiative and Pratham Books' Storyweaver are intended for children and most of them are vetted or human-verified we do not explicitly check for offensive content.

Our language identification models, by design, are meant to provide an alternative to training resource-hungry large-scale multilingual models that require a lot of training data. Such models are inaccessible to many researchers since they require access to specialized computing hardware. Our models are built with sustainability and equity in mind, and can be trained in a matter of minutes on CPU on standard laptops.

## Acknowledgments

This work was generously supported by the National Endowment for the Humanities under award PR-276810-21, by the National Science Foundation under award IIS-2125466, and by a Sponsored Research Award from Meta. Computational resources for experiments were provided by the Office of Research Computing at George Mason University (URL: https://orc.gmu.edu) and funded in part by grants from the National Science Foundation (Awards Number 1625039 and 2018631).

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

## A  Reproducibility

In this section, we outline how to reproduce the different aspects our work. Data collection, data preprocessing, machine translation experiments and evaluation, and language identification experiments have been completed in a manner that is fully reproducible.

### A.1  Data Curation

All data can be replicated and reproduced through `code/data-collection`. Intermediate preprocessing steps can be applied through `code/preprocessing`, merged through `code/merging`, and summary statistics be produced through `code/summary-stats`. Data paths are set up so that any retrieved, preprocessed, merged data is located in `data/`.

### A.2  Language ID

`code/language-id/` contains the relveant scripts to replicate all language identification experiments, training, model architecture, and results. Relevant language identification data is decoupled from the code directory and is located in `data/language-id`.

### A.3  Machine Translation

Our machine translation experiments are performed using publicly available code from https://github.com/mahfuzibnalam/large-scale_MT_African_languages. To produce results regarding novel translation directions enabled by our data, please refer to `code/new_lang_pairs`. Table A.1 shows the phylogeny configuration we use to fine-tune the MT system.

## B  Supplementary Machine Translation Benchmarks

On the following pages, we report the aggregate evaluation results of our MT models on the FLORES200 devtest of 176 languages (BLEU, CHRF++, spBLEU). We also report BLEU, CHRF++, and spBLEU for baseline, language-fine, and family-fine scores for all language pairs we perform machine translation experiments for (African focus languages from the WMT tasks' focus languages)

| Family | Genus (Group) | Language |
|---|---|---|
| Indo-European | Germanic | English |
| | | Afrikaans |
| | Romance | French |
| Afro-Asiatic | Hausa | Hausa |
| | Amharic | Amharic |
| | Cushitic | Oromo |
| | Cushitic | Somali |
| Nilo-Saharan | Luo | Luo |
| Atlantic | Wolof | Wolof |
| | Fula | Nigerian Fulfulde |
| Volta-Niger | Igboid | Igbo |
| | Yoruboid | Yoruba |
| Bantu | Bangi | Lingala |
| | Shona | Shona |
| | Nyasa | Chichewa |
| | Umbundu | Umbundu |
| | Sotho-Tswana | Tswana |
| | | Northern Sotho |
| | Nguni-Tsonga | Zulu |
| | | Xhosa |
| | | Swati |
| | | Xitsonga |
| | Northeast-Bantu | Kamba |
| | | Swahili |
| | | Kinyarwanda |
| | | Luganda |

Table A.1: This table highlights the three-tiered phylogeny-informed tree structure we use for `L-Fine` and `F-Fine` models. Other than the high-resource languages English, Afrikaans, and French, all other evaluated languages are from Africa and cover 5 different language families.

| Metrics | Models | $\text{Avg}_{\text{ALL}}$ | $\text{Avg}_{\text{X}\rightarrow\text{ENG}}$ | $\text{Avg}_{\text{ENG}\rightarrow\text{X}}$ | $\text{Avg}_{\text{AFRICAN}\rightarrow\text{AFRICAN}}$ | $\text{Avg}_{\text{Y}\rightarrow\text{FRA}}$ | $\text{Avg}_{\text{FRA}\rightarrow\text{Y}}$ |
|---|---|---|---|---|---|---|---|
| BLEU | Baseline | 9.59 | 17.84 | 10.56 | 7.83 | 13.42 | 9.79 |
| | L-Fine | 16.57 | 28.55 | 13.68 | 15.28 | 19.16 | 14.24 |
| | F-Fine | 21.52 | 33.77 | 21.79 | 20.01 | 23.99 | 17.28 |
| CHRF++ | Baseline | 29.59 | 35.47 | 30.88 | 28.05 | 32.54 | 31.25 |
| | L-Fine | 37.04 | 45.54 | 35.24 | 35.89 | 39.11 | 36.82 |
| | F-Fine | 41.33 | 49.83 | 41.28 | 40.18 | 43.27 | 39.28 |
| spBLEU | Baseline | 11.87 | 18.79 | 13.20 | 10.19 | 15.64 | 12.55 |
| | L-Fine | 19.52 | 30.38 | 17.46 | 18.21 | 21.93 | 17.86 |
| | F-Fine | **24.93** | **35.66** | **25.26** | **23.58** | **27.06** | **21.36** |

Table B.1: Evaluation results on our test set of 176 language directions. $\text{Avg}_{\text{X}\rightarrow\text{ENG}}$ denotes the average score of directions between other languages and English. $\text{Avg}_{\text{ENG}\rightarrow\text{X}}$ denotes the average score of directions between English and other languages. $\text{Avg}_{\text{AFRICAN}\rightarrow\text{AFRICAN}}$ denotes the average score of directions between African languages to other African languages. $\text{Avg}_{\text{Y}\rightarrow\text{FRA}}$ denotes the average score of directions between other languages and French. $\text{Avg}_{\text{FRA}\rightarrow\text{Y}}$ denotes the average score of directions between French and other languages. $\text{Avg}_{\text{ALL}}$ denotes the average result of all translation directions.

| Metrics | Models | $\text{Avg}_{\text{ALL}}$ | $\text{Avg}_{\text{X}\rightarrow\text{ENG}}$ | $\text{Avg}_{\text{ENG}\rightarrow\text{X}}$ | $\text{Avg}_{\text{AFRICAN}\rightarrow\text{AFRICAN}}$ | $\text{Avg}_{\text{Y}\rightarrow\text{FRA}}$ | $\text{Avg}_{\text{FRA}\rightarrow\text{Y}}$ |
|---|---|---|---|---|---|---|---|
| BLEU | Baseline | 14.01 | 28.21 | 13.68 | 23.62 | 11.66 | 11.23 |
| | L-Fine | 12.91 | 25.98 | 14.19 | 22.21 | 10.98 | 10.05 |
| | F-Fine | 12.25 | 25.15 | 13.99 | 20.98 | 10.88 | 9.34 |
| CHRF++ | Baseline | 39.16 | 49.54 | 37.16 | 46.24 | 38.03 | 37.29 |
| | L-Fine | 37.89 | 47.28 | 39.86 | 44.80 | 37.26 | 35.57 |
| | F-Fine | 37.03 | 46.57 | 39.64 | 43.68 | 37.41 | 34.50 |
| spBLEU | Baseline | 18.23 | 30.77 | 17.22 | 28.01 | 16.69 | 15.64 |
| | L-Fine | 17.01 | 28.23 | 18.91 | 26.46 | 15.75 | 14.22 |
| | F-Fine | 16.15 | 27.41 | 18.69 | 25.25 | 15.69 | 13.21 |

Table B.2: Evaluation results on FLORES200 devtest of 176 language directions. $\text{Avg}_{\text{X}\rightarrow\text{ENG}}$ denotes the average score of directions between other languages and English. $\text{Avg}_{\text{ENG}\rightarrow\text{X}}$ denotes the average score of directions between English and other languages. $\text{Avg}_{\text{AFRICAN}\rightarrow\text{AFRICAN}}$ denotes the average score of directions between African languages to other African languages. $\text{Avg}_{\text{Y}\rightarrow\text{FRA}}$ denotes the average score of directions between other languages and French. $\text{Avg}_{\text{FRA}\rightarrow\text{Y}}$ denotes the average score of directions between French and other languages. $\text{Avg}_{\text{ALL}}$ denotes the average result of all translation directions.

| | BLEU | | | CHRF++ | | | spBLEU | | |
|---|---|---|---|---|---|---|---|---|---|
| | **Baseline** | **L-Fine** | **F-Fine** | **Baseline** | **L-Fine** | **F-Fine** | **Baseline** | **L-Fine** | **F-Fine** |
| LUG-ENG | 10.1 | 17.3 | 23.4 | 25.1 | 33.8 | 39.6 | 10.9 | 18.2 | 24.5 |
| YOR-ENG | 16.2 | 22.2 | 26.3 | 33.6 | 40.1 | 43.8 | 16.3 | 22.1 | 26.5 |
| HAU-ENG | 15.9 | 21.5 | 24.6 | 33.6 | 39 | 40.7 | 17.2 | 22.7 | 25.7 |
| AMH-ENG | 21.4 | 27.8 | 30.1 | 41.3 | 45.8 | 47.6 | 23.2 | 29 | 31.3 |
| SWA-ENG | 22.5 | 26.6 | 33.3 | 41.1 | 44.1 | 49.4 | 23.4 | 27.5 | 34.5 |
| IBO-ENG | 15.1 | 20.9 | 22.8 | 34 | 40.2 | 41.8 | 15.8 | 22.3 | 24.1 |
| NYA-ENG | 17.9 | 25.2 | 33.4 | 37.3 | 44.5 | 51.4 | 18.7 | 25.5 | 35.2 |
| ORM-ENG | 7.9 | 11.3 | 15 | 24.1 | 30 | 33.3 | 9.4 | 12.6 | 16.5 |
| NSO-ENG | 22.5 | 46.7 | 53.5 | 39.5 | 61.7 | 66.5 | 23.4 | 51.1 | 57.1 |
| XHO-ENG | 21.9 | 36.5 | 42.2 | 39 | 52.1 | 56.9 | 22.4 | 38.2 | 44.7 |
| TSO-ENG | 20.5 | 48.3 | 57.1 | 37.1 | 62 | 68.7 | 21.1 | 52.1 | 60.3 |
| KIN-ENG | 13.4 | 22.5 | 29.1 | 31.1 | 40.3 | 45.8 | 13.9 | 23.1 | 30 |
| KAM-ENG | 5.3 | 8 | 11.7 | 21.8 | 24.6 | 29.9 | 6.1 | 8.9 | 12.9 |
| ZUL-ENG | 19.5 | 34.8 | 41.3 | 37.9 | 51.9 | 57.1 | 20.7 | 39.1 | 45.5 |
| SSW-ENG | 17.4 | 41.9 | 48.8 | 33.8 | 56.6 | 61.9 | 17.9 | 45.9 | 52.5 |
| AFR-ENG | 38 | 45.3 | 47.7 | 57.2 | 61.9 | 62.9 | 40.3 | 47.7 | 49.3 |
| ENG-SWA | 16.8 | 18.1 | 24.5 | 41.4 | 43 | 47 | 20 | 22.1 | 27.8 |
| ENG-IBO | 13.6 | 17.6 | 20.9 | 33.4 | 37.4 | 39.7 | 17.1 | 21 | 24.3 |
| ENG-NYA | 10.4 | 11.7 | 15.6 | 33.5 | 36.3 | 39.4 | 12 | 14.3 | 18.4 |
| ENG-ORM | 1.1 | 3.1 | 2.9 | 14.1 | 18.5 | 17.7 | 1.7 | 4.4 | 4.4 |
| ENG-NSO | 19.5 | 25.1 | 43.2 | 37.7 | 43.6 | 59.9 | 19.8 | 26 | 46.4 |
| ENG-TSO | 16 | 23.6 | 44.6 | 36.8 | 43.8 | 61.2 | 16.9 | 25.6 | 47.8 |
| ENG-KIN | 5.3 | 7.8 | 10.1 | 26.3 | 31.1 | 33.5 | 7.7 | 11.2 | 13.9 |
| ENG-KAM | 1.6 | 2.1 | 2.9 | 16.3 | 18.6 | 20.2 | 2.4 | 3.1 | 4.3 |
| ENG-ZUL | 9 | 11.2 | 27.8 | 36.4 | 38.9 | 49.3 | 16 | 18.7 | 32.9 |
| ENG-SSW | 5.7 | 9.9 | 36.2 | 29.3 | 36.8 | 53.5 | 10.2 | 16.7 | 37.9 |
| ENG-AFR | 33.1 | 35.1 | 40.3 | 52.8 | 54.6 | 57.7 | 35.6 | 37.4 | 42.4 |
| ENG-XHO | 4.5 | 12.4 | 28.2 | 26.5 | 37.7 | 48.2 | 8.1 | 18.1 | 31.5 |
| ENG-LUG | 4.2 | 5.9 | 8.1 | 25.7 | 29.8 | 31.9 | 7 | 9.8 | 12.3 |
| ENG-YOR | 13.3 | 15.8 | 20.4 | 30.9 | 32.5 | 35.7 | 14.8 | 17.3 | 20.8 |
| ENG-HAU | 10.4 | 12.4 | 15 | 31.3 | 34.9 | 37.8 | 8.7 | 14.2 | 17.8 |
| ENG-AMH | 4.4 | 7 | 7.9 | 21.7 | 26.3 | 27.7 | 13.2 | 19.4 | 21.3 |
| FRA-SWA | 8.9 | 12.9 | 17.6 | 34 | 39 | 42.6 | 11.1 | 16 | 21.3 |
| FRA-KIN | 5 | 7.6 | 9.9 | 28.4 | 32 | 34.5 | 7.6 | 11.6 | 14.5 |
| FRA-HAU | 6.5 | 8.9 | 11.8 | 29.5 | 33.6 | 36.1 | 7.8 | 10.7 | 14 |
| FRA-NSO | 15.2 | 23.9 | 29.2 | 34.2 | 45.4 | 49.9 | 16 | 26.3 | 32.3 |
| FRA-AMH | 2.9 | 5 | 5.9 | 17.6 | 21.7 | 21.9 | 10.2 | 14.3 | 15.8 |
| FRA-XHO | 9.2 | 12.2 | 15.8 | 35.8 | 40.1 | 42.1 | 14.5 | 19 | 22.2 |
| FRA-ZUL | 6.9 | 9.4 | 12.7 | 35.3 | 38.8 | 41.1 | 13 | 16.3 | 20 |
| FRA-LUG | 2.3 | 5.3 | 7.3 | 21.9 | 29.2 | 31.3 | 4.3 | 8.7 | 11.3 |
| FRA-IBO | 13 | 18.1 | 20.5 | 32 | 37.1 | 38.6 | 16 | 21.2 | 23.5 |
| FRA-AFR | 27.1 | 30.1 | 31.6 | 47.3 | 49 | 50.3 | 28.5 | 31.3 | 33.3 |
| FRA-NYA | 9 | 11.8 | 14.9 | 33.2 | 36.3 | 39.1 | 10.6 | 13.8 | 17.4 |
| FRA-SSW | 5.7 | 12.4 | 17.7 | 29.8 | 39.8 | 44.2 | 9.7 | 18.2 | 24.1 |

|         | BLEU     |        |        | CHRF++   |        |        | spBLEU   |        |        |
|---------|----------|--------|--------|----------|--------|--------|----------|--------|--------|
|         | Baseline | L-Fine | F-Fine | Baseline | L-Fine | F-Fine | Baseline | L-Fine | F-Fine |
| FRA-YOR | 9.6      | 14.1   | 15.5   | 23.9     | 28     | 29.5   | 9.9      | 13.8   | 16     |
| FRA-TSO | 15.8     | 27.6   | 31.5   | 34.6     | 45.5   | 48.7   | 16.5     | 28.9   | 33.3   |
| HAU-FRA | 9        | 13.9   | 17.3   | 26.2     | 32.5   | 35.8   | 11.4     | 16.8   | 20.7   |
| NSO-FRA | 15.2     | 23.3   | 29.9   | 35.8     | 44.5   | 49.7   | 18       | 26.3   | 33     |
| AMH-FRA | 12       | 16.7   | 19.1   | 30.2     | 35.6   | 37.4   | 13.4     | 18.9   | 21.2   |
| XHO-FRA | 16.7     | 22.1   | 26.6   | 37.1     | 42.9   | 46.2   | 18.2     | 23.9   | 28.8   |
| ZUL-FRA | 14.4     | 19.9   | 24.5   | 36.2     | 41.5   | 45.5   | 16.9     | 23.1   | 28.3   |
| LUG-FRA | 6.4      | 14.1   | 19.5   | 21.6     | 32.6   | 38.5   | 8.4      | 16.5   | 22.2   |
| IBO-FRA | 10.9     | 15.1   | 17.6   | 31.5     | 36.8   | 38.9   | 13.6     | 18.5   | 21.5   |
| AFR-FRA | 26       | 28.5   | 31.5   | 47.8     | 50.3   | 51.8   | 29.6     | 32.4   | 35.2   |
| NYA-FRA | 14       | 20.4   | 27.2   | 32.7     | 39.5   | 45.6   | 15.7     | 22.5   | 29.6   |
| SSW-FRA | 14.7     | 21.4   | 28.8   | 34.2     | 42     | 48     | 16.7     | 24.4   | 32.3   |
| YOR-FRA | 10.4     | 16     | 18.2   | 30.1     | 35.6   | 38     | 11.9     | 18.2   | 21     |
| TSO-FRA | 18.3     | 26     | 35.5   | 36.1     | 43.9   | 51.3   | 20.4     | 29     | 38.5   |
| SWA-FRA | 12       | 15.3   | 19.2   | 30.5     | 34.5   | 38.4   | 14.1     | 18     | 22.2   |
| KIN-FRA | 7.9      | 15.5   | 21     | 25.5     | 35.4   | 40.7   | 10.6     | 18.5   | 24.3   |
| TSO-SWA | 13.4     | 17.7   | 27.2   | 37.6     | 41.8   | 48.8   | 17.4     | 22.8   | 31.9   |
| SSW-TSO | 13.7     | 35.6   | 40.9   | 33.5     | 54.3   | 58.3   | 13.8     | 38.9   | 44.1   |
| AMH-KIN | 2.6      | 5.8    | 7.1    | 21       | 26.2   | 27.3   | 4.6      | 8.2    | 9.7    |
| TSO-NYA | 6.4      | 8.7    | 17     | 31.5     | 36.6   | 43.7   | 9.4      | 12.5   | 22.7   |
| TSO-NSO | 18.6     | 37     | 43.6   | 36.4     | 55.4   | 60.2   | 17.8     | 40.7   | 46.8   |
| NSO-KIN | 4.8      | 9.2    | 14.2   | 27.9     | 33.9   | 37.7   | 8.2      | 13.7   | 19.7   |
| YOR-IBO | 13.3     | 24.9   | 26.7   | 28.4     | 39.3   | 41.5   | 15.6     | 26.2   | 28.4   |
| SSW-SWA | 8.6      | 13.2   | 23.2   | 34.1     | 39.8   | 48.4   | 11.7     | 16.8   | 28.6   |
| NYA-SWA | 9.8      | 14     | 22.9   | 35.1     | 38.6   | 45.6   | 12.2     | 17.2   | 27.4   |
| YOR-SWA | 10.4     | 17.8   | 21.5   | 30.3     | 39.1   | 42.8   | 12.2     | 18.5   | 23.4   |
| SSW-NSO | 13.8     | 38.2   | 44.9   | 31.6     | 54.9   | 60.4   | 13.8     | 40.6   | 47.5   |
| SSW-NYA | 5.9      | 9.3    | 17.7   | 28.1     | 35.5   | 42.1   | 7.9      | 13.5   | 22.5   |
| AFR-SWA | 11.8     | 15.5   | 23.7   | 38.7     | 41.7   | 46     | 16.1     | 19.8   | 28.6   |
| XHO-TSO | 13.1     | 35.6   | 42.1   | 34       | 53.7   | 59.7   | 14.1     | 37.1   | 44.4   |
| LUG-NYA | 2.3      | 7.4    | 12.2   | 20.2     | 30     | 34.2   | 4        | 9.7    | 15.3   |
| AMH-AFR | 10.4     | 16.9   | 19.5   | 28.6     | 34.1   | 36.8   | 11.9     | 18.1   | 21     |
| LUG-NSO | 5.7      | 14.6   | 18.8   | 20.7     | 31.9   | 36.1   | 5.5      | 14.8   | 19.3   |
| NSO-AFR | 20.6     | 30.9   | 43.3   | 40       | 50.9   | 58.9   | 21.9     | 33.4   | 45.2   |
| HAU-KIN | 2.6      | 5      | 6.1    | 21.7     | 26.8   | 27.6   | 5.2      | 8.6    | 9.5    |
| IBO-SWA | 14.1     | 19.9   | 26.7   | 35       | 39.8   | 44.6   | 15.6     | 21     | 27.6   |
| AMH-ZUL | 3.1      | 6.7    | 7.9    | 26.2     | 31.2   | 32.5   | 7.5      | 11.8   | 13.5   |
| LUG-SWA | 6.3      | 11.4   | 17.7   | 26.5     | 34.8   | 40     | 8.2      | 14.7   | 21.8   |
| LUG-IBO | 4.8      | 13.7   | 17.3   | 18       | 28.6   | 33     | 6.8      | 15.6   | 19.7   |
| NSO-ZUL | 7.8      | 23.9   | 27     | 34.3     | 45.6   | 48.8   | 13.6     | 27     | 30.9   |
| ZUL-SWA | 9.5      | 13.7   | 20.9   | 34.9     | 37.9   | 43.4   | 13.3     | 17.7   | 25.8   |
| XHO-SWA | 13       | 16.6   | 23.4   | 37.1     | 40.2   | 45.3   | 16.2     | 20.1   | 27.4   |
| LUG-XHO | 3.7      | 7.7    | 12.4   | 22.1     | 29.7   | 34.9   | 5.5      | 11.2   | 16.8   |
| XHO-NSO | 15.1     | 32.8   | 40.5   | 34.5     | 51.6   | 57.9   | 15.5     | 35.5   | 43.5   |

| | BLEU | | | CHRF++ | | | spBLEU | | |
|---|---|---|---|---|---|---|---|---|---|
| | **Baseline** | **L-Fine** | **F-Fine** | **Baseline** | **L-Fine** | **F-Fine** | **Baseline** | **L-Fine** | **F-Fine** |
| ZUL-NYA | 6.1 | 7.7 | 11.2 | 31.2 | 33.6 | 36.8 | 8.8 | 10.9 | 15.5 |
| KAM-SWA | 4.1 | 4.4 | 7.2 | 22.3 | 21.2 | 25.8 | 5.8 | 5.8 | 9.5 |
| XHO-NYA | 6.8 | 10 | 18 | 31.6 | 36.4 | 43.2 | 9.7 | 13.5 | 22.6 |
| TSO-KIN | 4.4 | 7.1 | 13 | 26.2 | 29.4 | 36.1 | 6.8 | 10.6 | 18.7 |
| NSO-NYA | 5.6 | 8.1 | 15 | 30.2 | 35.6 | 40.5 | 8.1 | 12.2 | 19 |
| LUG-AFR | 9.7 | 14.2 | 22.3 | 27.9 | 32.7 | 39.9 | 10.6 | 15.1 | 23.4 |
| AMH-ORM | 0.8 | 2 | 3.1 | 14.3 | 19.8 | 22.9 | 1.3 | 3.8 | 5.1 |
| AMH-SWA | 5.7 | 14.8 | 18.3 | 27 | 38 | 41.1 | 7.8 | 19 | 22.2 |
| SWA-KIN | 4.3 | 5.3 | 7.3 | 23.3 | 24.2 | 27.5 | 6.3 | 7.5 | 10.5 |
| LUG-ZUL | 3.3 | 5.7 | 8.6 | 22.9 | 28.1 | 32.2 | 5.7 | 9.3 | 13.3 |
| NSO-SWA | 13.5 | 19.5 | 27.9 | 36.8 | 42.9 | 48.4 | 16.9 | 24.3 | 32.4 |
| XHO-SSW | 4.5 | 36.2 | 40.9 | 27.4 | 51.9 | 56.9 | 8.9 | 37.4 | 43.2 |
| SSW-KIN | 5.3 | 10 | 15.3 | 26.9 | 33.5 | 39 | 8.2 | 13.9 | 20.7 |
| NYA-KIN | 4.9 | 9.2 | 13.3 | 26.7 | 31.8 | 36.9 | 7 | 12.4 | 18 |
| YOR-LUG | 2.1 | 5.7 | 7.9 | 18.7 | 26.8 | 27.9 | 3 | 7.5 | 9.7 |
| XHO-ZUL | 9.1 | 32.4 | 36.1 | 33.9 | 49.7 | 53.5 | 14.9 | 33.5 | 38.4 |
| XHO-AFR | 19.7 | 24.5 | 33.9 | 39.8 | 44.4 | 51.8 | 21.4 | 26.1 | 36 |
| ZUL-AFR | 18 | 22.4 | 30.2 | 38.4 | 42.8 | 49.1 | 20.1 | 24.4 | 32.8 |
| TSO-ZUL | 6.3 | 39.4 | 41.8 | 31.4 | 53.7 | 56 | 11.2 | 38 | 41.5 |
| AFR-KIN | 5 | 7.6 | 11.3 | 29.7 | 32.9 | 37.6 | 7.9 | 11.6 | 16.6 |
| HAU-SWA | 8.3 | 10.5 | 13.5 | 29.3 | 32.6 | 34.5 | 10 | 13.6 | 16.2 |
| ORM-SWA | 5.6 | 8.4 | 9.6 | 23.4 | 27.7 | 28.1 | 6.4 | 10.9 | 11.6 |
| TSO-AFR | 19.9 | 25.9 | 36.3 | 38.4 | 44.3 | 52.8 | 21.1 | 27.1 | 38.1 |
| LUG-KIN | 1.1 | 6.3 | 7.8 | 15 | 27.2 | 29.5 | 2 | 8.7 | 11 |
| ZUL-KIN | 4.8 | 6.8 | 10.4 | 27.4 | 30.6 | 33.4 | 7 | 10.1 | 14.6 |
| SSW-ZUL | 8.5 | 31 | 33.3 | 33.8 | 50.6 | 52.2 | 13.9 | 33.3 | 36 |
| XHO-KIN | 4.8 | 7.6 | 11.1 | 27.9 | 31.3 | 35.2 | 7.6 | 11.6 | 16.3 |
| LUG-AMH | 1 | 2.4 | 4.7 | 8.5 | 13.6 | 16.7 | 3.2 | 8.2 | 11.4 |
| SSW-AFR | 17.1 | 23.5 | 36.3 | 35 | 42.1 | 51.8 | 18.2 | 25.1 | 38.1 |
| NYA-AFR | 15.4 | 20.8 | 29.7 | 33 | 38.6 | 45.9 | 16.7 | 21.8 | 31.3 |
| SWA-TSO | 13.8 | 21.1 | 29.7 | 35.6 | 41.6 | 49.5 | 15.3 | 22.7 | 32.4 |
| TSO-SSW | 6.3 | 31.1 | 35 | 31 | 50 | 54.2 | 10.5 | 33.1 | 38.3 |
| KIN-AMH | 0.7 | 1.8 | 2.9 | 9.7 | 14.8 | 16.9 | 3.5 | 8.4 | 11 |
| NYA-TSO | 13.3 | 24.7 | 31.5 | 33.2 | 45.1 | 50.6 | 14.2 | 27.2 | 34.3 |
| NSO-TSO | 15.9 | 37.5 | 42.2 | 36.2 | 56.5 | 60.3 | 16.3 | 41.3 | 46.2 |
| KIN-NSO | 6.4 | 17.8 | 24.2 | 22.7 | 36.1 | 41.8 | 6.1 | 17.9 | 25 |
| IBO-YOR | 12.9 | 17.1 | 18.3 | 26.2 | 29.9 | 30.3 | 13 | 16.7 | 17.4 |
| SWA-SSW | 3.9 | 6.5 | 11.4 | 23.5 | 28.7 | 34.4 | 5.8 | 10.2 | 16.1 |
| SWA-NYA | 6.5 | 7.4 | 11.4 | 27.6 | 29 | 33.4 | 8.3 | 9.8 | 14.3 |
| SWA-YOR | 9.3 | 13.3 | 15.7 | 21.3 | 24.7 | 28 | 9.4 | 13.8 | 15.5 |
| NSO-SSW | 5.7 | 36.3 | 40.2 | 29.7 | 52.8 | 56.6 | 10 | 36.9 | 41.4 |
| NYA-SSW | 5.1 | 10.5 | 19.2 | 27.3 | 36.3 | 44.6 | 7.7 | 16 | 25.9 |
| SWA-AFR | 15.2 | 18.4 | 25 | 34 | 36.2 | 41.9 | 16.8 | 19.5 | 26.8 |
| TSO-XHO | 8.8 | 33.3 | 37.2 | 34.9 | 51.3 | 54.9 | 13.8 | 33.2 | 38.4 |

| | BLEU | | | CHRF++ | | | spBLEU | | |
|---|---|---|---|---|---|---|---|---|---|
| | **Baseline** | **L-Fine** | **F-Fine** | **Baseline** | **L-Fine** | **F-Fine** | **Baseline** | **L-Fine** | **F-Fine** |
| NYA-LUG | 2.5 | 4.7 | 8.1 | 22.9 | 26.4 | 29.7 | 4.5 | 7.5 | 11.3 |
| AFR-AMH | 2.6 | 3.8 | 5.3 | 17.4 | 21.1 | 23.8 | 9.2 | 12.7 | 15.6 |
| NSO-LUG | 2.6 | 4 | 5.1 | 23.3 | 27.4 | 28.1 | 4.6 | 7 | 8.9 |
| AFR-NSO | 18.6 | 28.5 | 34.7 | 38.3 | 48.7 | 53 | 18.8 | 29.5 | 36 |
| KIN-HAU | 6.3 | 8.3 | 11.1 | 26 | 28.4 | 32.8 | 7.6 | 10.3 | 13.9 |
| SWA-IBO | 18.4 | 23.5 | 27.3 | 32.5 | 37.4 | 41 | 20.1 | 24.7 | 28.5 |
| ZUL-AMH | 2 | 4.2 | 5.4 | 15.9 | 19.4 | 22.8 | 7.4 | 11.8 | 15.8 |
| SWA-LUG | 3.5 | 5.1 | 7.2 | 23.2 | 25.2 | 28.3 | 5.8 | 7.8 | 10.4 |
| IBO-LUG | 1.6 | 5.5 | 7.7 | 20.3 | 28.2 | 29.9 | 2.9 | 8.2 | 10.8 |
| ZUL-NSO | 15.7 | 29.1 | 34.6 | 36 | 47.8 | 52.8 | 15.9 | 30.6 | 36.6 |
| SWA-ZUL | 7.2 | 8.3 | 11.6 | 30.2 | 31.5 | 34.9 | 10.9 | 12.5 | 16.6 |
| SWA-XHO | 8.8 | 9.8 | 15.7 | 33.9 | 35.1 | 40.6 | 13.4 | 14.8 | 21.6 |
| XHO-LUG | 3.7 | 5.4 | 8.1 | 26.6 | 30.4 | 31.9 | 5.6 | 8.6 | 11.9 |
| NSO-XHO | 8 | 29.9 | 33.9 | 34.1 | 49.2 | 53 | 13.5 | 32.1 | 36.7 |
| NYA-ZUL | 6.9 | 8.8 | 14 | 31.3 | 34.5 | 39.4 | 11.2 | 14.2 | 20.1 |
| SWA-KAM | 1.5 | 1.3 | 1.8 | 16.2 | 16.3 | 17.8 | 2.3 | 2 | 3 |
| NYA-XHO | 8.1 | 11.3 | 20.5 | 33.2 | 38.9 | 46.1 | 12.8 | 18.2 | 27.5 |
| KIN-TSO | 6.2 | 15.8 | 22.2 | 22.9 | 34 | 39.8 | 6.6 | 17 | 24.1 |
| NYA-NSO | 14.9 | 24.2 | 32.9 | 34 | 44.2 | 50.5 | 15.1 | 25.8 | 34.5 |
| AFR-LUG | 4 | 5.4 | 7 | 28.3 | 31.2 | 32.6 | 6.9 | 9.3 | 11.4 |
| ORM-AMH | 1 | 1.8 | 2.8 | 10.5 | 15.7 | 16.3 | 3.2 | 9.9 | 10.8 |
| SWA-AMH | 2.1 | 4.4 | 5.8 | 14.8 | 19.4 | 22.5 | 7.4 | 12.9 | 16.9 |
| KIN-SWA | 6.3 | 11.2 | 18.7 | 28.4 | 34.2 | 40.1 | 9 | 14.8 | 23.2 |
| ZUL-LUG | 3.2 | 4 | 5.2 | 24 | 25.2 | 26.3 | 6 | 7.3 | 8.2 |
| SWA-NSO | 15.2 | 20.1 | 29.4 | 33.9 | 39.6 | 46.8 | 15.2 | 21.3 | 31 |
| SSW-XHO | 6.1 | 36.4 | 40.5 | 29.1 | 51.4 | 55.2 | 10.6 | 36.6 | 41.3 |
| KIN-SSW | 1.9 | 10 | 16.9 | 19.9 | 33.6 | 40.8 | 4 | 15.7 | 24.2 |
| KIN-NYA | 2.4 | 6.2 | 11.4 | 22.1 | 31.5 | 36.7 | 3.4 | 8.4 | 15.3 |
| LUG-YOR | 3.2 | 8 | 13.1 | 12.6 | 19.1 | 25.2 | 3.3 | 8.4 | 13.5 |
| ZUL-XHO | 8.9 | 32.6 | 35.7 | 34.4 | 49.8 | 52.7 | 14.5 | 33.4 | 37.2 |
| AFR-XHO | 10.1 | 12.3 | 16.2 | 37.7 | 40.3 | 43.2 | 16.3 | 19.2 | 23.4 |
| AFR-ZUL | 9.5 | 11.2 | 15.1 | 36.5 | 38.8 | 42 | 15.3 | 18 | 22.3 |
| ZUL-TSO | 13 | 39.6 | 44 | 34 | 56.6 | 59.9 | 13.8 | 42.1 | 46.6 |
| KIN-AFR | 10.9 | 15.3 | 23.6 | 29 | 33.6 | 41.3 | 11.3 | 16 | 25.3 |
| SWA-HAU | 7.8 | 10.7 | 13 | 30.4 | 31.1 | 35.1 | 9.3 | 12 | 14.8 |
| SWA-ORM | 2 | 2.2 | 3.3 | 17.9 | 20.1 | 22.2 | 3.3 | 3.6 | 5.2 |
| AFR-TSO | 18.2 | 24.9 | 29.5 | 39.6 | 48.2 | 52.3 | 19.6 | 28 | 33 |
| KIN-LUG | 0.3 | 4.6 | 6.2 | 14.6 | 26.6 | 28.4 | 1.4 | 8 | 9.6 |
| KIN-ZUL | 3.9 | 6.6 | 9.5 | 24.2 | 30.3 | 33.2 | 6.6 | 11.2 | 14.8 |
| ZUL-SSW | 6.4 | 30.4 | 33.8 | 33 | 51.2 | 53.9 | 12.4 | 32.8 | 37.2 |
| KIN-XHO | 3.4 | 8.7 | 12.1 | 24 | 31.3 | 34.3 | 5.8 | 12.8 | 16.8 |
| AMH-LUG | 1.4 | 4.2 | 4.6 | 18.4 | 24.7 | 24.9 | 3.1 | 6.2 | 7.2 |
| AFR-SSW | 6.4 | 11.9 | 18 | 28.9 | 37.8 | 43.4 | 10 | 18 | 25.1 |
| AFR-NYA | 6.4 | 8.4 | 13 | 30.1 | 33.5 | 37.7 | 9.6 | 12.1 | 17.5 |

Table B.3: Results of all language pairs on our test set

|  | BLEU | | | CHRF++ | | | spBLEU | | |
|---|---|---|---|---|---|---|---|---|---|
|  | **Baseline** | **L-Fine** | **F-Fine** | **Baseline** | **L-Fine** | **F-Fine** | **Baseline** | **L-Fine** | **F-Fine** |
| LUG-ENG | 16.1 | 15.5 | 15.5 | 36.6 | 36 | 36.2 | 18.3 | 17.5 | 17.7 |
| YOR-ENG | 16.7 | 15.6 | 16.3 | 38.6 | 37.2 | 38.2 | 18.9 | 17.7 | 18.4 |
| HAU-ENG | 27.8 | 27.2 | 25.9 | 50.2 | 49.1 | 48 | 31.1 | 29.2 | 27.9 |
| AMH-ENG | 31.4 | 27.8 | 27.1 | 55.5 | 52.1 | 51.3 | 34 | 30.1 | 29.3 |
| SWA-ENG | 41.6 | 34.2 | 36.6 | 62.5 | 56.5 | 58.4 | 43.5 | 36.2 | 38.5 |
| IBO-ENG | 25.6 | 24 | 23.8 | 48.3 | 45.3 | 45.4 | 28.6 | 26.3 | 26.1 |
| NYA-ENG | 25.2 | 23.3 | 22.9 | 47.8 | 45.5 | 44.9 | 28.7 | 26.3 | 25.7 |
| ORM-ENG | 13.2 | 11.7 | 10.3 | 34.6 | 32.2 | 29.6 | 14.5 | 12.7 | 11.1 |
| NSO-ENG | 34.6 | 32.5 | 29.5 | 55 | 53 | 50.2 | 36.8 | 34.5 | 31.5 |
| XHO-ENG | 35.1 | 33.4 | 31.7 | 56.2 | 55.1 | 53.3 | 37.9 | 36.2 | 34.3 |
| TSO-ENG | 28.1 | 25.8 | 24.4 | 49.6 | 47.2 | 46.2 | 30.8 | 28.2 | 26.9 |
| KIN-ENG | 28.1 | 24.2 | 23.3 | 50 | 46.4 | 45.5 | 30.2 | 26.2 | 25.4 |
| KAM-ENG | 9.5 | 10 | 9.1 | 28.3 | 28.2 | 28.7 | 12.2 | 12.3 | 12.2 |
| ZUL-ENG | 35.8 | 32.3 | 31.1 | 57.5 | 53.8 | 52.8 | 38.7 | 34.6 | 33.3 |
| SSW-ENG | 26.1 | 25.6 | 24.1 | 47.6 | 47.1 | 45.7 | 28.5 | 27.9 | 26.3 |
| AFR-ENG | 56.5 | 52.6 | 50.8 | 74.4 | 71.8 | 70.7 | 59.6 | 55.7 | 53.9 |
| ENG-SWA | 33.8 | 30.8 | 29.8 | 59.4 | 57.3 | 56.4 | 38 | 35.3 | 34.4 |
| ENG-IBO | 15.8 | 16.1 | 16.3 | 39.5 | 40.1 | 40.2 | 18.6 | 19 | 19.2 |
| ENG-NYA | 14.2 | 13.8 | 13.4 | 44.5 | 44.5 | 43.6 | 18.1 | 17.7 | 16.9 |
| ENG-ORM | 1.3 | 1 | 0.7 | 18.2 | 17.1 | 15.4 | 2.4 | 1.7 | 1.2 |
| ENG-NSO | 23.1 | 19.1 | 19.4 | 47.9 | 44.9 | 45.7 | 24.4 | 21 | 21.5 |
| ENG-TSO | 16.4 | 15.6 | 16.8 | 43.7 | 42.5 | 43.9 | 19.6 | 18.2 | 19.4 |
| ENG-KIN | 12.5 | 11 | 11.3 | 37.9 | 38.1 | 38.2 | 15.9 | 14.5 | 14.7 |
| ENG-KAM | 2.8 | 3.9 | 4.2 | 19.3 | 22.4 | 22.8 | 3.8 | 5.4 | 5.6 |
| ENG-ZUL | 16.1 | 15.3 | 14.3 | 50.2 | 49.5 | 48.5 | 27.2 | 26.2 | 24.7 |
| ENG-SSW | 7.6 | 7 | 7 | 39 | 38.6 | 38.8 | 14.7 | 14.6 | 14.3 |
| ENG-AFR | 40.4 | 37.5 | 35.8 | 65.7 | 63.6 | 62.4 | 46.1 | 43.4 | 41.7 |
| ENG-XHO | 1.4 | 12.8 | 13.9 | 15.7 | 46.6 | 47.6 | 3.5 | 22.5 | 23.6 |
| ENG-LUG | 5.4 | 5.8 | 6.1 | 29.8 | 30.9 | 31.2 | 7 | 8 | 8.5 |
| ENG-YOR | 3.3 | 3.3 | 3.2 | 19.5 | 19.2 | 19.2 | 5.1 | 4.6 | 4.6 |
| ENG-HAU | 13.1 | 22.3 | 20.7 | 27.7 | 46.9 | 45.6 | 4.5 | 24.2 | 23.3 |
| ENG-AMH | 11.6 | 11.8 | 10.9 | 36.6 | 35.5 | 34.7 | 26.8 | 26.2 | 25.4 |
| FRA-SWA | 23.6 | 20.5 | 20.1 | 50.9 | 48.6 | 47.5 | 28.1 | 25.1 | 24.3 |
| FRA-KIN | 9.6 | 8.6 | 9.1 | 36.4 | 34.6 | 35.6 | 13.4 | 11.8 | 12.2 |
| FRA-HAU | 15.4 | 15.3 | 15.1 | 41 | 40.5 | 40.7 | 18.1 | 17.8 | 17.7 |
| FRA-NSO | 12.8 | 12.3 | 13.1 | 38.6 | 38 | 39.4 | 14.9 | 14.4 | 15.3 |
| FRA-AMH | 8.5 | 6.9 | 6.5 | 31.7 | 28.5 | 27.8 | 22.2 | 19.6 | 19.1 |
| FRA-XHO | 10.3 | 9.1 | 9 | 43.1 | 41.1 | 41.2 | 19.4 | 17 | 17.3 |
| FRA-ZUL | 11.1 | 10 | 9.6 | 44.9 | 43.5 | 42.9 | 21.6 | 19.9 | 19.1 |
| FRA-LUG | 2.2 | 4.3 | 4.2 | 22 | 28 | 28.7 | 3 | 6.4 | 6.6 |
| FRA-IBO | 13.1 | 12.1 | 12.5 | 36.9 | 35.6 | 36.4 | 16.1 | 14.9 | 15.3 |
| FRA-AFR | 26.7 | 24.6 | 22.6 | 54.9 | 52.6 | 50.8 | 32.7 | 30.1 | 28.2 |
| FRA-NYA | 11.6 | 10.1 | 10.1 | 42.2 | 39.7 | 39.6 | 15.6 | 13.4 | 13.5 |
| FRA-SSW | 4.9 | 4.8 | 4.9 | 33.7 | 34.2 | 35.1 | 11.3 | 11.2 | 11.4 |

Table B.4: Results of all language pairs on our test set

| | BLEU | | | CHRF++ | | | spBLEU | | |
| --- | --- | --- | --- | --- | --- | --- | --- | --- | --- |
| | **Baseline** | **L-Fine** | **F-Fine** | **Baseline** | **L-Fine** | **F-Fine** | **Baseline** | **L-Fine** | **F-Fine** |
| FRA-YOR | 2.4 | 3.2 | 3 | 18.2 | 18.5 | 18.7 | 3.6 | 4.8 | 4.7 |
| FRA-TSO | 11 | 11.9 | 12.5 | 37.9 | 38.2 | 39.4 | 13.6 | 14.1 | 14.9 |
| HAU-FRA | 23.2 | 21.8 | 21.2 | 45.7 | 44.1 | 43.6 | 27.4 | 25.8 | 25.2 |
| NSO-FRA | 24.4 | 23.2 | 20.8 | 46.7 | 45.5 | 43.6 | 29 | 27.3 | 25.1 |
| AMH-FRA | 25.2 | 22.7 | 22.6 | 49.3 | 47.3 | 46.7 | 29.9 | 27.4 | 27.1 |
| XHO-FRA | 26.6 | 25.9 | 23.4 | 49.2 | 48.4 | 46 | 31.3 | 30.2 | 27.9 |
| ZUL-FRA | 28.1 | 25.3 | 22.7 | 51 | 48.4 | 46.1 | 32.5 | 29.6 | 27.1 |
| LUG-FRA | 13.1 | 13.6 | 12.5 | 33.8 | 34.3 | 33.6 | 16.2 | 16.6 | 15.6 |
| IBO-FRA | 20.2 | 18.8 | 18.1 | 43.1 | 41 | 40.7 | 24.2 | 22.5 | 22.4 |
| AFR-FRA | 37.9 | 37.1 | 35.8 | 60.8 | 59.9 | 58.9 | 43.6 | 42.7 | 41.5 |
| NYA-FRA | 20.5 | 19.7 | 18.5 | 44 | 42.5 | 41.2 | 25.4 | 24.3 | 23 |
| SSW-FRA | 19.7 | 20.4 | 18.3 | 41.9 | 43.1 | 40.9 | 24.1 | 24.5 | 22.4 |
| YOR-FRA | 15 | 13.5 | 13.5 | 37 | 35.4 | 35.3 | 18.9 | 17.5 | 17.5 |
| TSO-FRA | 22.4 | 20.9 | 19 | 44.8 | 43.5 | 41.5 | 26.7 | 25.2 | 22.8 |
| SWA-FRA | 31.7 | 27.3 | 27.7 | 54.4 | 50.4 | 50.8 | 36.1 | 31.9 | 32.2 |
| KIN-FRA | 22.7 | 20.7 | 19.6 | 45.7 | 43.4 | 42.6 | 26.8 | 24.9 | 23.7 |
| TSO-SWA | 19.3 | 16.3 | 13.8 | 45.7 | 42.8 | 39.3 | 22.9 | 20 | 17.1 |
| SSW-TSO | 12.1 | 13 | 12.4 | 38.6 | 39.4 | 38.2 | 15.1 | 15.5 | 14.6 |
| AMH-KIN | 8.2 | 6.7 | 6.5 | 35.1 | 32 | 31.4 | 11.6 | 9.5 | 8.9 |
| TSO-NYA | 10.3 | 9.9 | 8.3 | 39.2 | 38.4 | 35.1 | 13.8 | 13.2 | 11.2 |
| TSO-NSO | 17.3 | 15.5 | 14.1 | 42 | 40.6 | 39.2 | 18.9 | 17.3 | 15.9 |
| NSO-KIN | 9.6 | 9.4 | 7.9 | 35 | 34.6 | 31.9 | 12.9 | 12.4 | 10.3 |
| YOR-IBO | 8.4 | 7.9 | 8.2 | 29.7 | 29.2 | 29.4 | 11.1 | 10.6 | 10.9 |
| SSW-SWA | 17.4 | 16.4 | 13.1 | 43.5 | 42.6 | 38.6 | 20.6 | 19.5 | 16.2 |
| NYA-SWA | 17.8 | 15.1 | 13.3 | 44.9 | 41.4 | 39 | 21.6 | 18.9 | 16.7 |
| YOR-SWA | 11.9 | 9.4 | 9.9 | 37.6 | 33.4 | 34.2 | 14.6 | 11.9 | 12.4 |
| SSW-NSO | 16.1 | 14.9 | 13.8 | 40.8 | 39.9 | 38.1 | 17.7 | 16.7 | 15.3 |
| SSW-NYA | 9.2 | 10.2 | 8.3 | 37.4 | 38.5 | 35.2 | 12.5 | 13.1 | 11 |
| AFR-SWA | 27.6 | 24.2 | 20.9 | 54.8 | 51.5 | 48.4 | 32.1 | 28.7 | 25.7 |
| XHO-TSO | 13.8 | 13.7 | 13.7 | 40.5 | 40 | 40.1 | 16.8 | 16.4 | 16.2 |
| LUG-NYA | 7.2 | 7 | 6 | 32.5 | 32.4 | 30.4 | 9.4 | 9.4 | 8.1 |
| AMH-AFR | 19.4 | 17.3 | 16.3 | 46.5 | 44.1 | 43.1 | 23.3 | 20.8 | 19.6 |
| LUG-NSO | 9.9 | 10.7 | 9.5 | 32 | 33.7 | 32.8 | 10.9 | 12.1 | 11 |
| NSO-AFR | 20.9 | 18.8 | 16.8 | 45.8 | 43.4 | 41.1 | 24.2 | 21.3 | 19.2 |
| HAU-KIN | 10.1 | 8.4 | 7.6 | 36.5 | 32.7 | 31.5 | 13.7 | 10.8 | 9.9 |
| IBO-SWA | 16.6 | 15 | 14.7 | 44.4 | 40.5 | 40.7 | 20.8 | 18.1 | 17.7 |
| AMH-ZUL | 9 | 8.3 | 7.5 | 42.1 | 40.5 | 39.4 | 18.1 | 16.7 | 15.5 |
| LUG-SWA | 11.9 | 9.9 | 8.7 | 36.7 | 33.9 | 32.3 | 14.2 | 12.2 | 10.9 |
| LUG-IBO | 7.5 | 8 | 7.1 | 26.6 | 28.1 | 27.4 | 9.8 | 10.3 | 9.8 |
| NSO-ZUL | 12.7 | 10.8 | 9.3 | 44.9 | 42.5 | 40.2 | 22 | 19.7 | 17.4 |
| ZUL-SWA | 25 | 20.9 | 17.7 | 51.6 | 47.2 | 43.1 | 29 | 24.6 | 21 |
| XHO-SWA | 22.5 | 20.6 | 17.8 | 49.4 | 47.3 | 43.5 | 26.6 | 24.7 | 21.4 |
| LUG-XHO | 5.1 | 4.6 | 4.5 | 31.8 | 31.1 | 30.7 | 10.3 | 9.8 | 8.9 |
| XHO-NSO | 18.2 | 17.5 | 16.1 | 43.2 | 42.2 | 40.7 | 19.7 | 18.8 | 17.6 |

| | BLEU | | | CHRF++ | | | spBLEU | | |
|---|---|---|---|---|---|---|---|---|---|
| | **Baseline** | **L-Fine** | **F-Fine** | **Baseline** | **L-Fine** | **F-Fine** | **Baseline** | **L-Fine** | **F-Fine** |
| ZUL-NYA | 12.8 | 11.6 | 9.1 | 42.9 | 40.8 | 36.7 | 16.6 | 15 | 12.2 |
| KAM-SWA | 8.4 | 7.1 | 5.6 | 30.3 | 27.5 | 25.6 | 10.5 | 9.2 | 7.3 |
| XHO-NYA | 12.3 | 11.7 | 10.2 | 42.1 | 40.8 | 37.7 | 16 | 15.2 | 13.1 |
| TSO-KIN | 10 | 9 | 6.8 | 36.3 | 34.2 | 30.6 | 14.1 | 12.3 | 9.2 |
| NSO-NYA | 11.1 | 10.6 | 9.4 | 39.6 | 38.9 | 36.5 | 14.3 | 13.5 | 12 |
| LUG-AFR | 11.6 | 10.6 | 9.9 | 34 | 32.9 | 31.8 | 14 | 13 | 11.9 |
| AMH-ORM | 1.2 | 0.8 | 0.9 | 19.5 | 16.7 | 17.9 | 2.5 | 1.5 | 1.6 |
| AMH-SWA | 20.2 | 17.1 | 16.4 | 48.4 | 44.8 | 43.3 | 24.3 | 20.9 | 19.7 |
| SWA-KIN | 12.2 | 7.4 | 7.9 | 39.9 | 32.1 | 32.9 | 16.1 | 10.2 | 10.6 |
| LUG-ZUL | 5.4 | 5.3 | 4.7 | 33.1 | 32.6 | 31.1 | 11.9 | 11.4 | 10.2 |
| NSO-SWA | 21.7 | 19 | 15.3 | 48.2 | 45.1 | 40.6 | 25.2 | 22.3 | 18.4 |
| XHO-SSW | 7.2 | 7 | 6.3 | 37.7 | 37.2 | 35.6 | 14.3 | 14.1 | 12.7 |
| SSW-KIN | 7.9 | 8.7 | 6.6 | 32.4 | 33.5 | 30 | 10.7 | 11.3 | 8.6 |
| NYA-KIN | 9.2 | 7.7 | 7.1 | 35 | 32.2 | 31 | 12.5 | 10.7 | 9.5 |
| YOR-LUG | 3.6 | 3.9 | 3.3 | 25.2 | 24.9 | 24.6 | 4.9 | 5.6 | 5.2 |
| XHO-ZUL | 12.9 | 12.1 | 11.1 | 45.8 | 44.9 | 42.8 | 23.2 | 22.3 | 20.1 |
| XHO-AFR | 20.7 | 19.8 | 17.2 | 46.7 | 45.3 | 42.7 | 24.8 | 23.3 | 20.5 |
| ZUL-AFR | 22.4 | 18.8 | 17.4 | 48.3 | 44.3 | 42.4 | 26.3 | 22.2 | 20.3 |
| TSO-ZUL | 10.5 | 9.4 | 8.4 | 42.7 | 41 | 39.1 | 20.3 | 18.1 | 16.3 |
| AFR-KIN | 11.5 | 9 | 8.8 | 38.9 | 35.2 | 34.9 | 15.9 | 12.2 | 12 |
| HAU-SWA | 20.9 | 17.1 | 15.2 | 47.4 | 42.7 | 40.9 | 24.2 | 20.3 | 18.3 |
| ORM-SWA | 10.5 | 7.2 | 6.3 | 34.7 | 28.8 | 26.5 | 12.2 | 8.6 | 7.3 |
| TSO-AFR | 17.1 | 16 | 14 | 42.2 | 41 | 38.4 | 20.9 | 19.3 | 17.1 |
| LUG-KIN | 2.9 | 6.3 | 5.3 | 19.9 | 29 | 27.5 | 4.2 | 8.5 | 7.1 |
| ZUL-KIN | 10.9 | 9.2 | 7.4 | 37.9 | 34.5 | 31.9 | 14.8 | 12.4 | 10 |
| SSW-ZUL | 11.1 | 10.2 | 9.1 | 43 | 42.2 | 40.4 | 20.5 | 19.2 | 17.5 |
| XHO-KIN | 10.9 | 9.8 | 8.2 | 37.1 | 35.3 | 32.4 | 14.4 | 12.9 | 10.5 |
| LUG-AMH | 3.5 | 2.6 | 2.4 | 18.1 | 17.2 | 16.5 | 10.7 | 10 | 9.1 |
| SSW-AFR | 16 | 15.6 | 13.2 | 41.3 | 40.4 | 37.7 | 19.7 | 18.6 | 16 |
| NYA-AFR | 16.4 | 14.7 | 13.1 | 42.4 | 39.8 | 37.7 | 20.7 | 18.2 | 16.4 |
| SWA-TSO | 14.8 | 12.1 | 13.5 | 41.5 | 37 | 39.5 | 17.7 | 13.9 | 15.4 |
| TSO-SSW | 6.6 | 5.9 | 5.2 | 36.8 | 35.4 | 33.4 | 13.1 | 12.2 | 10.6 |
| KIN-AMH | 6 | 4 | 4.1 | 25.7 | 21.9 | 22 | 16.4 | 13.5 | 13.4 |
| NYA-TSO | 11.7 | 10.7 | 10.7 | 37 | 36.1 | 36.3 | 14.5 | 13.5 | 13.3 |
| NSO-TSO | 12.8 | 13.9 | 14 | 39.4 | 39.7 | 39.8 | 15.3 | 16.5 | 16.3 |
| KIN-NSO | 14.6 | 12.1 | 12.3 | 39.1 | 36.7 | 36.7 | 16.2 | 14.1 | 13.9 |
| IBO-YOR | 2.3 | 2.9 | 2.5 | 17.5 | 18 | 17.4 | 3.8 | 5.2 | 4.2 |
| SWA-SSW | 5.9 | 4.3 | 4.7 | 36.2 | 31.9 | 34.1 | 12.4 | 9.5 | 10.5 |
| SWA-NYA | 12.4 | 9.4 | 10 | 43.6 | 38.6 | 39.3 | 16.5 | 12.7 | 13.3 |
| SWA-YOR | 2.7 | 3.7 | 3.1 | 18.4 | 18.8 | 18.5 | 3.8 | 6.6 | 4.5 |
| NSO-SSW | 7.1 | 6.4 | 6.1 | 37.2 | 35.8 | 35.1 | 13 | 12.4 | 12 |
| NYA-SSW | 4.7 | 4.6 | 4.7 | 33 | 33 | 32.6 | 10.4 | 10.6 | 10 |
| SWA-AFR | 25.3 | 20 | 19.9 | 51.8 | 46.2 | 45.9 | 29.1 | 23.5 | 23.2 |
| TSO-XHO | 9.2 | 8.6 | 8 | 40.1 | 39.1 | 38.1 | 16.9 | 15.6 | 15.1 |

| | BLEU | | | CHRF++ | | | spBLEU | | |
|---|---|---|---|---|---|---|---|---|---|
| | **Baseline** | **L-Fine** | **F-Fine** | **Baseline** | **L-Fine** | **F-Fine** | **Baseline** | **L-Fine** | **F-Fine** |
| NYA-LUG | 3.2 | 4.6 | 4.6 | 24.2 | 27.4 | 26.9 | 4.5 | 6.9 | 6.6 |
| AFR-AMH | 9.3 | 7.6 | 8.2 | 33.1 | 30 | 30.7 | 23.1 | 21.1 | 21.9 |
| NSO-LUG | 3.4 | 5.3 | 5.3 | 24.6 | 29.3 | 28.5 | 4.4 | 7.6 | 7.4 |
| AFR-NSO | 18.7 | 14.9 | 15.4 | 44.8 | 41.1 | 42 | 20.7 | 17.3 | 17.8 |
| KIN-HAU | 14.9 | 12.9 | 12.3 | 39.6 | 36.4 | 36.3 | 17.5 | 15.3 | 14.6 |
| SWA-IBO | 14.3 | 11.8 | 12.7 | 38.3 | 34.9 | 36.4 | 17.2 | 14.9 | 15.7 |
| ZUL-AMH | 7.7 | 6.5 | 5.7 | 30.3 | 27.6 | 25.5 | 20.7 | 18.7 | 16.8 |
| SWA-LUG | 4.5 | 4.2 | 4.8 | 29.1 | 27.3 | 27.8 | 6.1 | 6.2 | 6.6 |
| IBO-LUG | 3.1 | 4.1 | 3.9 | 24.6 | 26.8 | 26.4 | 4.3 | 6.1 | 6 |
| ZUL-NSO | 18.8 | 16.7 | 15.8 | 44.5 | 41.6 | 41.1 | 20.8 | 18.3 | 17.5 |
| SWA-ZUL | 13.1 | 9.8 | 10.1 | 47.3 | 42.3 | 42.7 | 24 | 18.7 | 18.9 |
| SWA-XHO | 11.1 | 7.9 | 9.3 | 44.5 | 39 | 40.8 | 20.2 | 15 | 16.6 |
| XHO-LUG | 4 | 5.7 | 5 | 26.7 | 29.5 | 27.6 | 5.4 | 7.9 | 6.6 |
| NSO-XHO | 10.3 | 9.2 | 8.7 | 41.9 | 40.2 | 38.8 | 17.4 | 16.5 | 15.8 |
| NYA-ZUL | 8.8 | 7.8 | 7.2 | 40.6 | 38.6 | 37.1 | 17.8 | 16.2 | 14.8 |
| SWA-KAM | 2.7 | 2.8 | 2.8 | 19.5 | 20.7 | 20 | 4 | 4.3 | 4.1 |
| NYA-XHO | 7.7 | 6.7 | 6.4 | 38.7 | 36.4 | 35.6 | 15.7 | 13.7 | 13.2 |
| KIN-TSO | 13.1 | 10.7 | 10.9 | 39.3 | 35.8 | 36.2 | 15.9 | 12.9 | 12.8 |
| NYA-NSO | 12.4 | 12.1 | 12.5 | 36.6 | 36.6 | 37.4 | 14.2 | 14.1 | 14.6 |
| AFR-LUG | 4.8 | 4.7 | 4.7 | 28.8 | 28.5 | 28.9 | 6.3 | 6.7 | 7.1 |
| ORM-AMH | 3.8 | 2.8 | 2.2 | 21.1 | 18.4 | 16.4 | 11.8 | 10 | 8.4 |
| SWA-AMH | 8.5 | 5.5 | 7 | 31.9 | 26.4 | 28.5 | 21.8 | 17.5 | 19.1 |
| KIN-SWA | 19.6 | 15.6 | 13.8 | 46.2 | 41.6 | 39.1 | 23.1 | 19 | 16.7 |
| ZUL-LUG | 3.6 | 5.3 | 4.7 | 26.1 | 28.7 | 27.2 | 5 | 7.6 | 6.5 |
| SWA-NSO | 17.4 | 14.4 | 15.1 | 43.1 | 39.4 | 40.4 | 19.1 | 16 | 16.7 |
| SSW-XHO | 8.9 | 8.9 | 7.8 | 39.8 | 39.7 | 37.5 | 16.3 | 16.5 | 14.8 |
| KIN-SSW | 5.1 | 4.6 | 3.9 | 34.1 | 31.7 | 31.4 | 11.3 | 9.8 | 9 |
| KIN-NYA | 10.4 | 9 | 8.4 | 39.9 | 37.2 | 35.2 | 14 | 12.1 | 10.9 |
| LUG-YOR | 2.7 | 3.1 | 2.8 | 16.1 | 17.2 | 16.5 | 4.8 | 5.8 | 5 |
| ZUL-XHO | 12 | 10.4 | 10 | 45.1 | 42.9 | 41.5 | 21.3 | 19.2 | 18.1 |
| AFR-XHO | 11.1 | 9.7 | 9.5 | 44.6 | 42.4 | 42.1 | 20.5 | 18.3 | 18.1 |
| AFR-ZUL | 13 | 11.4 | 10.7 | 47.2 | 45.1 | 44.6 | 23.9 | 21.6 | 20.8 |
| ZUL-TSO | 14.3 | 14.1 | 14 | 41.9 | 40 | 40.3 | 17.4 | 16.9 | 16.7 |
| KIN-AFR | 17.2 | 15.3 | 14 | 42.3 | 39.7 | 37.8 | 20.4 | 17.8 | 16.2 |
| SWA-HAU | 19.5 | 14.8 | 16.5 | 45.8 | 38.9 | 41.6 | 22.3 | 17.2 | 19 |
| SWA-ORM | 1.1 | 0.7 | 0.6 | 18.3 | 15 | 14.6 | 2.3 | 1.1 | 0.9 |
| AFR-TSO | 15.4 | 13.5 | 13.8 | 43.2 | 40.1 | 41.2 | 18.9 | 16.2 | 16.6 |
| KIN-LUG | 1.9 | 4.1 | 4.1 | 19.3 | 26.8 | 26.5 | 3.4 | 6.2 | 5.8 |
| KIN-ZUL | 9.8 | 8.2 | 7.3 | 41.5 | 39.1 | 37.2 | 18.6 | 16.2 | 14.4 |
| ZUL-SSW | 7.3 | 7.2 | 6.6 | 39.3 | 38.2 | 37.1 | 14.9 | 15.1 | 13.7 |
| KIN-XHO | 8.6 | 6.9 | 6.8 | 39.1 | 36.6 | 35.7 | 15.7 | 13.5 | 12.6 |
| AMH-LUG | 2.6 | 3.2 | 3 | 24.7 | 25.8 | 25.4 | 3.6 | 5 | 4.6 |
| AFR-SSW | 6.3 | 5.4 | 5 | 37.7 | 36.1 | 35.9 | 13.6 | 12.5 | 11.9 |
| AFR-NYA | 12.3 | 11 | 10.3 | 43.1 | 41.1 | 40 | 16.4 | 14.6 | 13.9 |

Table B.5: Results of all language pairs on FLORES200 devtest