# OpenReview forum: "LIMIT: Language Identification, Misidentification, and Translation using Hierarchical Models in 350+ Languages"
_EMNLP/2023/Conference — EMNLP 2023 Main_

### Official Review · Reviewer_g5ek · 2023-07-26

**Soundness:** 4

**Excitement:**

4: Strong: This paper deepens the understanding of some phenomenon or lowers the barriers to an existing research direction.

**Paper Topic And Main Contributions:**

This paper deals with improving language identification for low-resource languages. Its main contributions are the following:

1) A new corpus, MCS-350, which is derived from two collections of original-language and human-translated children's stories. The stories are covered under permissive licenses so they can be used easily in future work.

2) A machine translation benchmark based on the parallel natures of the above dataset. They train a baseline MT system based on the highest-performing publicly-available model from WMT 2022, and show they can improve MT performance for African languages by fine-tuning adaptor units in a hierarchy based on language family.

3) A new hierarchical modelling approach based on learned confusion patterns in an existing model. It works by observing which languages are most often confused by an existing model, then training new subunits to distinguish between these languages. They present some analysis showing their method improves F1 scores for previously unsupported and confused languages.

**Questions For The Authors:**

A) How do you check that the corpus has the correct language labels?

B) In table 4, which baseline is being used and how many language pairs are being averaged in each column?

C) How did you choose the hyperparameters in section 4.1, footnote 8?

**Reasons To Accept:**

1) The MCS-350 corpus has the potential to be a valuable resource for the NLP community because it covers a large number of previously under-supported languages. I particularly like that they have included language varieties (though I hope that the mappings to non-standard codes are explained carefully in the repository). In addition, children's stories are a novel domain for low-resource language data, since most existing data is either religious text or Wikipedia.

2) Their hierarchical approach to language identification with learned confusion patterns is a novel and interesting idea.

3) The machine translation benchmark could be a useful addition to research into low-resource machine translation

**Reasons To Reject:**

1) My main concern with this paper is the lack of information on how the authors ensured the data was in the correct language. They mention in section 2.1 that they remove empty stories or those which have majority English text (according to langdetect), which to me implies that the other language labels may not be faithful. If this work is to be used as a benchmark for work on low-resource languages, then it is particularly important to check that it is in the right language to avoid "representation washing" (see section 6, https://aclanthology.org/2022.tacl-1.4.pdf). This is especially the case for the lowest-resource languages mentioned in section 2.2: the string similarity matching technique mentioned is not perfect, and the authors should say what they did to ensure the labels were correct.

2) Section 3 on the machine translation benchmark is currently lacking in some key details. In particular I would like more detail about the dataset, including how much data there is for each language and how parallel it actually is (i.e. which pairs of languages have parallel data and how much is there for each). With respect to the analysis, it is not clear which baseline is being used: in section 3.2 they say they use the DeltaLM model with fine-tuning on 26 languages, but then on line 254 they mention an additional baseline without adaptors. In addition, the results in table 4 lack information needed to understand the performance of the model: for example, it is not clear how many language pairs' scores are being averaged in each column, two of the columns have the same heading (Avg_{X -> ENG}, and there is no indication on how scores vary across languages (e.g. standard deviation). Overall, I would like more information about the dataset and the cross-language performance so I can understand the quality of the dataset.

3) In section 4, there is no comparison with existing work, only with the authors' `root` model which does not support all the languages. A strong baseline would strengthen the analysis (e.g. train a fasttext model on the new dataset), and even if the baseline outperformed the model, there is still a contribution as the authors' method can be used to extend existing models. In a related point, the comparison of other models in table 6 is not fair, since the models all cover different languages so higher F1 score is likely a function of higher coverage. The authors should note how many languages each LID model covers.

**Reproducibility:**

4: Could mostly reproduce the results, but there may be some variation because of sample variance or minor variations in their interpretation of the protocol or method.

**Reviewer Confidence:**

5: Positive that my evaluation is correct. I read the paper very carefully and I am very familiar with related work.

**Typos Grammar Style And Presentation Improvements:**

- Line 61: missing an 's' in 'translation systems"
- Could you explain what phylogeny means in the text? I don't think it's common enough a term to leave it unexplained (or just use 'language family').
- The captions for tables 4 and 5 do not explain the contents of the tables enough (e.g. in table 5, it's not clear if these are results from FLORES or MCS-350). The captions should explain what's in the table without having to refer to the text.
- In Table 7, under FLORES-200 (root), you include 0.00 scores for some languages which are not covered.

---

> ### Author Rebuttal · Authors · 2023-08-25
>
> We would like to genuinely thank you for your detailed comments. All the concerns you raised are important and we have provided clarifications, statistics, and new experimental results below. We address all the points one by one:
>
> > A) How do you check that the corpus has the correct language labels?
>
> - Both African Storybooks Initiative and Pratham Storyweaver verify stories and language.
> - For *unverified* stories, we implemented English text detection (since translators often copy English text into their story first and then translate it sentence by sentence) and empty story deletion (abandoned translation projects).
> - We manually verified all extracted monolingual text after using string matching on bilingual stories.
> - We provide metadata for Verified and Unverified stories at the source level, and versions of the dataset at each stage of pre-processing in the dataset.
>
> Thank you for raising this point! We care about the data’s quality and fully agree that ‘representation washing’ must be avoided. Details about the data preprocessing, verification and quality inspection are present on the GitHub but we will incorporate them into the main text as well (Section 2, Appendix).
>
> ---
>
> > B) In table 4, which baseline is being used?
>
> “Baseline” in Table 4 refers to DeltaLM model finetuned on 26 languages without adapters (line 255). L-Fine and F-Fine models are also built on top of DeltaLM but include adapters based on language or phylogeny (lines 247-254)
> We will rewrite the relevant lines to avoid confusion. Thank you for flagging it.
>
> ---
>
> > Table 4 column heading: X->Eng typo
>
> You’re right. The sixth column’s header contains a typo and should be Eng->X (with column values 13.20, 17.46, 25.26). We can see that it is harder to translate out of English than into English. We will correct this typo in the camera-ready and double check all other tables. Thank you!
>
> ---
>
> > In Table 4, how many language pairs are being averaged in each column?
>
> Here are the statistics for Table 4 for language pairs. Note that it doesn't include all the language pairs from MCS-350 by design and focuses on those that are common with FLORES-200 (for fair out-of-domain evaluation).
> - `All`: 176 (88 unique pairs)
> - `Afri->Afri`: 116 (58 unique pairs)
> - `X->Eng`: 16
> - `Eng->X`: 16
> - `Y->Fra`: 14
> - `Fra->Y`: 14
>
> We will redesign Table 4 to include these counts and the standard deviations (below) so the data can be properly interpreted. Thank you for highlighting this!
>
> ---
>
> > In Table 4, there is no indication on how scores vary across languages (e.g. standard deviation)
>
> We share standard deviations for Table 4 below. We will add this to the Appendix as well.
>
> |          | All   | Afr->Afr | X->Eng | Eng->X | Y->Fra | Fra->Y |
> | -------- | ----- | -------- | ------ | ------ | ------ | ------ |
> | Baseline | 6.31  | 5.06     | 7.75   | 8.19   | 5.22   | 5.81   |
> | L-Fine   | 10.33 | 10.06    | 13.63  | 8.46   | 4.87   | 6.86   |
> | F-Fine   | 11.74 | 11.31    | 14.36  | 13.72  | 6.00   | 7.32   |
>
> ---
>
> > Section 3 -  which pairs of languages have parallel data and how much is there for each
>
> - Due to space constraints, we omit the Afr->Afr matrix here, but we share training data sizes for English/French and African languages in both FLORES and MCS-350. Test set sizes are 1000, except for kam and orm, where total data < 1000, and test set size is 500 (line 235)
> - Statistics about each language pair in MT experiments are available in the data folder and on GitHub. But, we will certainly incorporate them into the Appendix to provide a better picture of the dataset!
>
> | Lang | w/ eng | w/ fra |
> | ---- | ------ | ------ |
> | kin  | 1334   | 379    |
> | kam  | 97     | \-     |
> | zul  | 1828   | 392    |
> | ssw  | 984    | 115    |
> | afr  | 1746   | 769    |
> | swa  | 5379   | 2914   |
> | ibo  | 363    | 280    |
> | nya  | 917    | 598    |
> | orm  | 444    | \-     |
> | nso  | 1104   | 102    |
> | xho  | 1873   | 790    |
> | tso  | 782    | 476    |
> | lug  | 1478   | 779    |
> | yor  | 319    | 132    |
> | hau  | 42     | 135    |
> | amh  | 1401   | 514    |
>
> ---
>
> > C) How did you choose the hyperparameters in section 4.1, footnote 8?
>
> We explored several starting learning rates and settled on lr=0.5  due to the performance on dev set. FastText’s lrUpdate parameter, by default, reduces the starting learning rate gradually over iterations. We don’t optimize for embedding dimension and use fastText’s default, 100, for supervised training. We will edit lines 347-358 to make these choices more clear.
>
> ---
>
> > In Table 7, under FLORES-200 (root), you include 0.00 scores for some languages which are not covered.
>
> asm (Assamese) and yue (Yue Chinese) are both covered by FLORES-200. The root model received 0 F1 score on the FLORES test sentences for these 2 languages. We’ll include this in the Table’s caption to avoid confusion on seeing zeros.
>
> ---
>
> > Stronger baseline (training fastText model on new data) required
>
> - Based on your request, we have attached an extended Table 7 demonstrating empirically that our model fares much better than a multilingual fastText model (`multi`). In addition, we also trained a traditional hierarchical model (`group`) that predicts language family first, followed by language.
> - We will include these details and additional strong baselines to strengthen the evidence in favor of our proposed technique (Section 4 and Table 7). Thanks for pointing this out, we agree that a stronger baseline helps the argument!
>
> **Highlights:**
> - The traditional hierarchical approach (family-then-language) underperforms the multilingual model on both MCS-350 and FLORES-200.
> - Our proposed LID method, LIMIT, outperforms the large multilingual model and the traditional hierarchical model across domains.
> - Aggregate results are present on the last line of the extended table below.
>
> |          | MCS-350 |              |              |              | FLORES  |              |       |              |
> | -------- | ------- | ------------ | ------------ | ------------ | ------------ | ------------ | ----- | ------------ |
> | Language | root    | multi        | group        | LIMIT        | root         | multi        | group | LIMIT        |
> | guj      | 0.49    | 0.44         | 0.58         | **0.63** | 1            | 0.81         | 0.99  | 0.99         |
> | kfr      |         | 0.78         | **0.83** | 0.8          |              |              |       |              |
> | bhil     |         | **0.63** | 0.03         | 0.28         |              |              |       |              |
> |          |         |              |              |              |              |              |       |              |
> | amh      | 0.2     | 0.81         | 0.78         | **0.83** | 0.6          | 0.95         | 0.93  | **0.99** |
> | tir      | 0.56    | 0.93         | **0.94** | 0.85         | **0.99** | 0.96         | 0.95  | 0.95         |
> | stv      |         | 0            | 0.00         | 0            |              |              |       |              |
> |          |         |              |              |              |              |              |       |              |
> | ben      | 0.47    | 0.82         | 0.05         | **0.85** | 1            | 0.94         | 0.96  | 0.99         |
> | asm      |         | 0.63         | 0.89         | **0.89** | 0            | **0.98** | 0.96  | 0.66         |
> | cdz      |         | 0.57         | 0.87         | **0.93** |              |              |       |              |
> |          |         |              |              |              |              |              |       |              |
> | zho      | 0.61    | 0            | 0.00         | **0.68** | 0.99         | 0            | 0.01  | 0.99         |
> | yue      |         | 0            | 0.02         | **0.14** | 0            | 0            | 0     | 0            |
> |          |         |              |              |              |              |              |       |              |
> | tel      | 0.92    | 0.75         | 0.80         | **0.94** | 1            | 0.83         | 0.91  | 1            |
> | kfc      |         | 0.66         | 0.66         | 0.66         |              |              |       |              |
> |          |         |              |              |              |              |              |       |              |
> | kan      | 0.7     | 0.77         | 0.78         | **0.81** | 1            | 0.96         | 0.93  | 0.99         |
> | kfa      |         | 0.66         | **0.68** | 0.52         |              |              |       |              |
> |          |         |              |              |              |              |              |       |              |
> | tso      | 0.49    | 0.53         | 0.34         | **0.67** | 0.97         | 0.79         | 0.72  | 0.97         |
> | tsc      |         | 0.74         | 0.52         | **0.77** |              |              |       |              |
> |          |         |              |              |              |              |              |       |              |
> | dagaare  | 0.83    | 0.48         | 0.54         | **0.87** |              |              |       |              |
> | mzm      |         | 0.69         | **0.88** | 0.82         |              |              |       |              |
> |          |         |              |              |              |              |              |       |              |
> | kat      | 0.66    | 0.46         | 0.42         | **0.8**  | 1            | 0.72         | 0.7   | 1            |
> | bbl      |         | **0.82** | 0.66         | 0.66         |              |              |       |              |
> |          |         |              |              |              |              |              |       |              |
> | **Average**  | 0.29    | 0.56         | 0.47         | **0.68** | 0.77         | 0.70         | 0.73  | **0.86** |
>
> ---
>
> > Comparison in Table 6 - coverage vs. high F1
>
> We share an updated Table 6 below with the language counts, where we can see that Franc wins on both coverage and performance (caption, Table 6). This certainly provides more transparency to the comparison, so we thank you for your comment. We will include this updated table in the camera-ready to provide more information on the root model’s selection.
> - "Common" means languages in common with the LangID model and our new MCS-350 dataset.
> - "Coverage with LIMIT" indicates the number of languages that the root model and our model, together, will cover
>
> | Model     | F1           | Supported | Common      | Coverage with LIMIT |
> | --------- | ------------ | --------- | ----------- | -------------- |
> | CLD3      | 0.11         | 101       | 81          | 376             |
> | langid.py | 0.09         | 97        | 73          | 380             |
> | Franc     | **0.18** | **369** | **116** | **609**    |
> | fastText  | 0.1          | 176       | 117         | 415             |
> | HeLI-OTS  | 0.13         | 200       | 81          | 475            |
>
> We hope our responses and additional experiments address your soundness concerns (as reflected by the soundness score).

---

### Official Review · Reviewer_Ri3q · 2023-08-03

**Soundness:** 3

**Excitement:**

3: Ambivalent: It has merits (e.g., it reports state-of-the-art results, the idea is nice), but there are key weaknesses (e.g., it describes incremental work), and it can significantly benefit from another round of revision. However, I won't object to accepting it if my co-reviewers champion it.

**Paper Topic And Main Contributions:**

This very ambitious paper presents a corpus for 350+ languages, evaluates several off-the-shelf language identification methods on a test set they created, proposes a hierarchical language identification system added on top of an existing language identifier, and train new machine translation models and evaluate them on an existing dataset.

**Questions For The Authors:**

Q A: As the main failing from the part of the off-the-shelf language identifiers seems to be their coverage of the languages, have you taken a look at the identifier with the record number of languages (1,400)? It is available at https://sourceforge.net/projects/la-strings/ (http://www.cs.cmu.edu/~ralf/langid.html)

**Reasons To Accept:**

- The compiled dataset seems useful and should be published.
- Further evaluation of existing off-the-shelf language identifiers on new challenging datasets and tasks is important.
- The research in each of the areas presented in the paper shows promise and continuing with it is worthwhile.

**Reasons To Reject:**

- The reasoning to use LIMIT is not on solid ground. What is the reason to "avoid training large multilingual models"? In the LIMIT method, the models for the missing languages are calculated anyway and the method adds complexity on top of an existing system. For the existing generative language identification systems, you need only to add the models for the new languages and there is no need to re-train all the languages. At least it would be nice to see how the end results compare with another system with all the same languages. Currently, there is no meaningful comparison for the LIMIT language identification results. The method would have to be very bad for it not to outperform a system missing many of the languages in the test set (referring to Table 7).
- The evaluation setting for language identification is in-domain.

**Reproducibility:**

4: Could mostly reproduce the results, but there may be some variation because of sample variance or minor variations in their interpretation of the protocol or method.

**Reviewer Confidence:**

5: Positive that my evaluation is correct. I read the paper very carefully and I am very familiar with related work.

**Typos Grammar Style And Presentation Improvements:**

Could you use commas in at least Table 1 to mark the thousands? The caption of Table 1 says "with with".

The heading "Parallel Datasets" on line 441 is not good as not all introduced datasets are parallel?

---

> ### Author Rebuttal · Authors · 2023-08-25
>
> Thank you for your feedback! Below, we address each of your questions/concerns one by one:
>
> > The reasoning to use LIMIT is not on solid ground. What is the reason to "avoid training large multilingual models"?
>
> - To clarify, by “large multilingual models”, we refer to LID systems, not multilingual LLMs.
> - Low-resource languages simply do not have enough data to warrant training of large multilingual LID models (Section 2). Instead of a 1-level k-way classification with large k, hierarchical models break down the problem into a multi-level classification task. Previous work has shown this to be successful for identifying closely-related languages and for low-resource setups (lines 505-524).
> - Large models like AfroLID train on >4000 sentences/lang, while we are often working with <=1000 (lines 351-354). We mention in the paper that large multilingual models are resource-hungry, data-hungry, and inaccessible to train (591-599).
>
> Thanks for highlighting this point. We will incorporate this into the paper’s Introduction section around line 97 to better motivate our proposed model.
>
> ---
> > In the LIMIT method, the models for the missing languages are calculated anyway and the method adds complexity on top of an existing system.
>
> Models need to be calculated for missing languages and some complexity must be added to extend a system *when retraining is not an option* (due to lack of data, closed-source code, etc). LIMIT’s “complexity” is an intuitive hierarchical model that achieves improved LID quality across domains with very little data.
> ---
> > For the existing generative language identification systems, you need only to add the models for the new languages and there is no need to re-train all the languages.
>
> The premise of our paper is that when faced with low-resource settings, existing generative models are unfit. Such low-quality or domain-specific “coverage” creates a vicious cycle (293-305). Similarly, not needing to retrain all languages doesn’t imply good performance. See, for instance, the difference between's Franc’s performance on supported languages on FLORES and the children's stories domain (Table 7). We will include this clarification in Section 4’s motivation.
> ---
> > At least it would be nice to see how the end results compare with another system with all the same languages. Currently, there is no meaningful comparison for the LIMIT language identification results.
>
> - With a large number of classes, low-resource languages suffer the most due to class imbalance in large multilingual models (lines 474-478).
> - Based on your request, we have attached an extended Table 7 demonstrating empirically that our model fares much better than a large multilingual model (trained on all the languages and training data). In addition, we also trained a traditional hierarchical model that predicts family first, followed by language.
> - We will include these details and additional strong baselines to strengthen the evidence in favor of our proposed technique (Section 4 and Table 7)!
>
> Highlights:
> - The traditional hierarchical approach (language family first, then language) underperforms the large multilingual model on both MCS-350 and FLORES-200.
> - Our hierarchical model, LIMIT, outperforms the large multilingual and the group-first models across domains. Aggregate results are available on the last line of the table below.
>
> |          | MCS-350 |              |              |              | FLORES  |              |       |              |
> | -------- | ------- | ------------ | ------------ | ------------ | ------------ | ------------ | ----- | ------------ |
> | Language | root    | multi        | group        | LIMIT        | root         | multi        | group | LIMIT        |
> | guj      | 0.49    | 0.44         | 0.58         | **0.63** | 1            | 0.81         | 0.99  | 0.99         |
> | kfr      |         | 0.78         | **0.83** | 0.8          |              |              |       |              |
> | bhil     |         | **0.63** | 0.03         | 0.28         |              |              |       |              |
> |          |         |              |              |              |              |              |       |              |
> | amh      | 0.2     | 0.81         | 0.78         | **0.83** | 0.6          | 0.95         | 0.93  | **0.99** |
> | tir      | 0.56    | 0.93         | **0.94** | 0.85         | **0.99** | 0.96         | 0.95  | 0.95         |
> | stv      |         | 0            | 0.00         | 0            |              |              |       |              |
> |          |         |              |              |              |              |              |       |              |
> | ben      | 0.47    | 0.82         | 0.05         | **0.85** | 1            | 0.94         | 0.96  | 0.99         |
> | asm      |         | 0.63         | 0.89         | **0.89** | 0            | **0.98** | 0.96  | 0.66         |
> | cdz      |         | 0.57         | 0.87         | **0.93** |              |              |       |              |
> |          |         |              |              |              |              |              |       |              |
> | zho      | 0.61    | 0            | 0.00         | **0.68** | 0.99         | 0            | 0.01  | 0.99         |
> | yue      |         | 0            | 0.02         | **0.14** | 0            | 0            | 0     | 0            |
> |          |         |              |              |              |              |              |       |              |
> | tel      | 0.92    | 0.75         | 0.80         | **0.94** | 1            | 0.83         | 0.91  | 1            |
> | kfc      |         | 0.66         | 0.66         | 0.66         |              |              |       |              |
> |          |         |              |              |              |              |              |       |              |
> | kan      | 0.7     | 0.77         | 0.78         | **0.81** | 1            | 0.96         | 0.93  | 0.99         |
> | kfa      |         | 0.66         | **0.68** | 0.52         |              |              |       |              |
> |          |         |              |              |              |              |              |       |              |
> | tso      | 0.49    | 0.53         | 0.34         | **0.67** | 0.97         | 0.79         | 0.72  | 0.97         |
> | tsc      |         | 0.74         | 0.52         | **0.77** |              |              |       |              |
> |          |         |              |              |              |              |              |       |              |
> | dagaare  | 0.83    | 0.48         | 0.54         | **0.87** |              |              |       |              |
> | mzm      |         | 0.69         | **0.88** | 0.82         |              |              |       |              |
> |          |         |              |              |              |              |              |       |              |
> | kat      | 0.66    | 0.46         | 0.42         | **0.8**  | 1            | 0.72         | 0.7   | 1            |
> | bbl      |         | **0.82** | 0.66         | 0.66         |              |              |       |              |
> |          |         |              |              |              |              |              |       |              |
> | **Average**  | 0.29    | 0.56         | 0.47         | **0.68** | 0.77         | 0.70         | 0.73  | **0.86** |
>
> ---
>
> > The method would have to be very bad for it not to outperform a system missing many of the languages in the test set (referring to Table 7).
>
> For the root model, LIMIT also outperforms on the languages that were *not missing* across domains (Table 7), ex. Gujarati, Amharic, Bengali etc. Our method also outperforms a system that is trained on all the languages in the test set. See group-first and large multilingual models in the extended Table 7 above. We will highlight this in the paper when we analyze our system (Section 4.3-4.4)!
>
> ---
>
> > The evaluation setting for language identification is in-domain.
>
> Our submission already includes evaluation in a different domain i.e FLORES-200. Please refer to lines 20-21 (Abstract), 103, 336-339, 421-425, 544-546 for details. For analysis, refer to lines 427-439
>
> ---
>
> > Have you taken a look at the identifier with the record number of languages (1,400)?
>
> We assume that you’re referring to the 2014 paper (that trains richer embeddings with non-linear mappings). We have looked at it and we didn’t use it because:
> - No experiments with the technique in low-resource training setups (current training data is 2.5 million bytes per language)
> - There is no analysis of cross-domain impact of the mapping techniques
> - Newer embedding methods trained on larger corpora like fastText have since been released
>
> We will cite this paper in the Related Work section under “Language Identification” and mention our reasoning for not using the mapping technique on fastText embeddings. Thank you for pointing this out.
>
> ---
>
> > Reproducibility
>
> - We have attached all code and data in a ZIP folder. Footnote on Page 1 indicates that “data, code, and models will be shared on GitHub publicly under permissive licenses.”
> - Instructions to reproduce all experiments and data collection (219-220, 256-257) are already provided. In Appendix A (1027-1070), we have outlined the relevant directories/scripts as well. We will expand Appendix A to include more details about the code and data statistics to encourage reproducibility!

---

### Official Review · Reviewer_BgTR · 2023-08-03

**Soundness:** 4

**Excitement:**

4: Strong: This paper deepens the understanding of some phenomenon or lowers the barriers to an existing research direction.

**Paper Topic And Main Contributions:**

- This paper constructs MCS-350, a dataset of 50,000+ parallel children's stories from the African Storybooks Initiative and Storyweaver, in 350+ languages.

- In this paper, the authors present LIMIT, a misidentification-based hierarchical model designed to enhance the identification of low-resource languages using limited data

- This paper presents a machine translation benchmark that allows evaluation in more than 1400 new translation directions.

**Reasons To Accept:**

- The paper introduces MCS-350, a dataset of 50,000+ parallel children's stories in 350+ languages, offering a valuable resource for future research.

-  The LIMIT model proposed in the paper utilizes limited data effectively to improve language identification, providing a novel and practical approach.

**Reasons To Reject:**

- The authors' presentation of the Language (Mis)identification LIMIT could have been more detailed, and additional experimentation on the method could have strengthened their findings



**Reproducibility:**

5: Could easily reproduce the results.

**Reviewer Confidence:**

3: Pretty sure, but there's a chance I missed something. Although I have a good feel for this area in general, I did not carefully check the paper's details, e.g., the math, experimental design, or novelty.

---

> ### Author Rebuttal · Authors · 2023-08-25
>
> >Presentation of LIMIT could have been more detailed, and additional experimentation on the method could have strengthened their findings
>
> Thank you for your comments! Below, we share additional experimental results (extension of Table 7) as per your request. We trained 2 additional stronger baselines:
> - `multi`: a large fastText multilingual model with all 350+ languages
> - `group`: a traditional hierarchical model that predicts language family first, followed by language
>
> Highlights:
> - The traditional hierarchical model underforms the multilingual model on both MCS-350 and FLORES-200.
> - Our proposed method, LIMIT, outperforms the large multilingual model and the group-first baselines across domains.
> - Please see the last row of the table below for aggregate results.
>
> We will include these details to strengthen the evidence in favor of our proposed technique and the presentation! Thanks again for your feedback and scores.
>
>
> |          | MCS-350 |              |              |              | FLORES  |              |       |              |
> | -------- | ------- | ------------ | ------------ | ------------ | ------------ | ------------ | ----- | ------------ |
> | Language | root    | multi        | group        | LIMIT        | root         | multi        | group | LIMIT        |
> | guj      | 0.49    | 0.44         | 0.58         | **0.63** | 1            | 0.81         | 0.99  | 0.99         |
> | kfr      |         | 0.78         | **0.83** | 0.8          |              |              |       |              |
> | bhil     |         | **0.63** | 0.03         | 0.28         |              |              |       |              |
> |          |         |              |              |              |              |              |       |              |
> | amh      | 0.2     | 0.81         | 0.78         | **0.83** | 0.6          | 0.95         | 0.93  | **0.99** |
> | tir      | 0.56    | 0.93         | **0.94** | 0.85         | **0.99** | 0.96         | 0.95  | 0.95         |
> | stv      |         | 0            | 0.00         | 0            |              |              |       |              |
> |          |         |              |              |              |              |              |       |              |
> | ben      | 0.47    | 0.82         | 0.05         | **0.85** | 1            | 0.94         | 0.96  | 0.99         |
> | asm      |         | 0.63         | 0.89         | **0.89** | 0            | **0.98** | 0.96  | 0.66         |
> | cdz      |         | 0.57         | 0.87         | **0.93** |              |              |       |              |
> |          |         |              |              |              |              |              |       |              |
> | zho      | 0.61    | 0            | 0.00         | **0.68** | 0.99         | 0            | 0.01  | 0.99         |
> | yue      |         | 0            | 0.02         | **0.14** | 0            | 0            | 0     | 0            |
> |          |         |              |              |              |              |              |       |              |
> | tel      | 0.92    | 0.75         | 0.80         | **0.94** | 1            | 0.83         | 0.91  | 1            |
> | kfc      |         | 0.66         | 0.66         | 0.66         |              |              |       |              |
> |          |         |              |              |              |              |              |       |              |
> | kan      | 0.7     | 0.77         | 0.78         | **0.81** | 1            | 0.96         | 0.93  | 0.99         |
> | kfa      |         | 0.66         | **0.68** | 0.52         |              |              |       |              |
> |          |         |              |              |              |              |              |       |              |
> | tso      | 0.49    | 0.53         | 0.34         | **0.67** | 0.97         | 0.79         | 0.72  | 0.97         |
> | tsc      |         | 0.74         | 0.52         | **0.77** |              |              |       |              |
> |          |         |              |              |              |              |              |       |              |
> | dagaare  | 0.83    | 0.48         | 0.54         | **0.87** |              |              |       |              |
> | mzm      |         | 0.69         | **0.88** | 0.82         |              |              |       |              |
> |          |         |              |              |              |              |              |       |              |
> | kat      | 0.66    | 0.46         | 0.42         | **0.8**  | 1            | 0.72         | 0.7   | 1            |
> | bbl      |         | **0.82** | 0.66         | 0.66         |              |              |       |              |
> |          |         |              |              |              |              |              |       |              |
> | **Average**  | 0.29    | 0.56         | 0.47         | **0.68** | 0.77         | 0.70         | 0.73  | **0.86** |

---

### Meta-Review · Area_Chair_mBK2 · 2023-09-25

**Recommendation:** 4

**Metareview:**

Summary (adapted from reviewer Ri3q): This paper presents a corpus for 350+ languages, evaluates several off-the-shelf language identification methods on a test set they created, proposes a hierarchical language identification system added on top of an existing language identifier, and trains new machine translation models and evaluates them on an existing dataset.

All reviewers were positive regarding the value of the new corpus developed in this paper.

Reviewers raised some questions regarding the details of the experiments, the vast majority of which seem to be addressed well in the authors’ response. While the paper should be updated to reflect all the additional information provided during the discussion period, these are mostly clarifications and the addition of some useful but not necessarily essential baselines. The paper submitted for review was still sound, but these additions will enhance its soundness.

The submitted paper also did not clearly justify why the specific hierarchical model needed to be used, in comparison to other models. This was clarified in the authors’ response as well.

Overall, the corpus and the models represent significant contributions for the field of language ID for less-resourced languages.

---

### Decision · Program_Chairs · 2023-10-07

**Decision:**

Accept-Main

**Comment:**

Summary (adapted from reviewer Ri3q): This paper presents a corpus for 350+ languages, evaluates several off-the-shelf language identification methods on a test set they created, proposes a hierarchical language identification system added on top of an existing language identifier, and trains new machine translation models and evaluates them on an existing dataset.

All reviewers were positive regarding the value of the new corpus developed in this paper.

Reviewers raised some questions regarding the details of the experiments, the vast majority of which seem to be addressed well in the authors’ response. While the paper should be updated to reflect all the additional information provided during the discussion period, these are mostly clarifications and the addition of some useful but not necessarily essential baselines. The paper submitted for review was still sound, but these additions will enhance its soundness.

The submitted paper also did not clearly justify why the specific hierarchical model needed to be used, in comparison to other models. This was clarified in the authors’ response as well.

Overall, the corpus and the models represent significant contributions for the field of language ID for less-resourced languages.